# Auditing for Human Expertise

**Rohan Alur**
EECS
MIT
ralur@mit.edu

**Loren Laine**
School of Medicine
Yale
loren.laine@yale.edu

**Darrick K. Li**
School of Medicine
Yale
darrick.li@yale.edu

**Manish Raghavan**
Sloan, EECS
MIT
mragh@mit.edu

**Devavrat Shah**
EECS, IDSS, LIDS, SDSC
MIT
devavrat@mit.edu

**Dennis Shung**
School of Medicine
Yale
dennis.shung@yale.edu

## Abstract

High-stakes prediction tasks (e.g., patient diagnosis) are often handled by trained human experts. A common source of concern about automation in these settings is that experts may exercise intuition that is difficult to model and/or have access to information (e.g., conversations with a patient) that is simply unavailable to a would-be algorithm. This raises a natural question whether human experts *add value* which could not be captured by an algorithmic predictor.

We develop a statistical framework under which we can pose this question as a natural hypothesis test. Indeed, as our framework highlights, detecting human expertise is more subtle than simply comparing the accuracy of expert predictions to those made by a particular learning algorithm. Instead, we propose a simple procedure which tests whether expert predictions are statistically independent from the outcomes of interest after conditioning on the available inputs ('features'). A rejection of our test thus suggests that human experts may add value to *any* algorithm trained on the available data, and has direct implications for whether human-AI 'complementarity' is achievable in a given prediction task.

We highlight the utility of our procedure using admissions data collected from the emergency department of a large academic hospital system, where we show that physicians' admit/discharge decisions for patients with acute gastrointestinal bleeding (AGIB) appear to be incorporating information that is not available to a standard algorithmic screening tool. This is despite the fact that the screening tool is arguably *more* accurate than physicians' discretionary decisions, highlighting that – even absent normative concerns about accountability or interpretability – accuracy is insufficient to justify algorithmic automation.

## 1 Introduction

Progress in machine learning, and in algorithmic decision aids more generally, has raised the prospect that algorithms may complement or even automate human decision making in a wide variety of settings. If implemented carefully, these tools have the potential to improve accuracy, fairness, interpretability and consistency in many prediction and decision tasks. However, a primary challenge in nearly all such settings is that some of the relevant inputs – 'features,' in machine learning parlance – are difficult or even impossible to encode in a way that an algorithm can easily consume. For example, doctors use direct conversations with patients to inform their diagnoses, and sports franchises employ professional scouts to qualitatively assess prospective players. One can think of these experts as incorporating information which is practically difficult to provide to an algorithm, particularly as

37th Conference on Neural Information Processing Systems (NeurIPS 2023).

tabular data, or perhaps exercising judgment which is infeasible to replicate with a computational process. Either perspective presents a challenge when deploying predictive algorithmic tools, as any such model will necessarily fail to incorporate at least some of the information that a human expert might consider. Thus, as we seek to use algorithms to improve decision-making, we must answer the following question:

*For a given prediction task, do human experts add value which could not be captured by any algorithmic forecasting rule?*

The answer to this question has significant consequences: if experts are incorporating salient but hard-to-quantify information, we might attempt to somehow ensemble or combine the human and algorithmic predictions; this is commonly referred to as seeking 'complementarity' in the literature on human-machine interaction. On the other hand, if it appears that an expert is not extracting signal beyond whatever is contained in the available features, we might consider whether we can automate the prediction task entirely, or at least reduce the degree to which a human may override algorithmic recommendations.

At this stage it is worth asking – why not simply compare the prediction accuracy of a human expert to that of a particular predictive algorithm? If the human expert performs better than a competing algorithm, we might say the expert adds value which is not captured by the algorithm. However, as the example presented next illustrates, it is possible for the expert to incorporate information that could not be captured by *any* learning algorithm – even when the expert substantially underperforms a particular algorithm trained to accomplish the same task. Indeed, this is not just a hypothetical: a large body of prior literature (see e.g. Agrawal (2019) for a comprehensive overview) suggests that humans reliably underperform even simple statistical models, and in Section 5, we find exactly this dynamic in real-world patient triage data. Nonetheless, as we highlight next, humans may still add valuable information in a given forecasting task.

**An illustration: experts may add information despite poor predictions.** Let $Y$ denote the outcome of interest and let $X, U$ be features that drive the outcome. Specifically, let

$$Y = X + U + \epsilon_1, \tag{1}$$

where $\epsilon_1$ is some exogenous noise. For the purposes of this stylized example, we'll assume that $X, U, \epsilon_1$ are all zero mean and pairwise independent random variables. Suppose the human expert can observe both $X$ and $U$, but only $X$ is made available to a predictive algorithm. An algorithm tasked with minimizing squared error might then seek to precisely estimate $\mathbb{E}[Y|X] = X$. In contrast, the expert may instead use simpler heuristics to construct an estimate which can be modeled as

$$\hat{Y} = \text{sign}(X) + \text{sign}(U) + \epsilon_2, \tag{2}$$

where $\epsilon_2$ is independent zero-mean noise, and can be thought of as modeling idiosyncracies in the expert's cognitive process[1]. As discussed in detail in Appendix F, there exist natural distributions over $(X, U, \epsilon_1, \epsilon_2)$ such that the algorithm performs substantially better than the expert in terms of predictive accuracy. In fact, we show that there exist natural distributions where the algorithm outperforms the expert even under any linear post-processing of $\hat{Y}$ (e.g., to correct for expert predictions which are highly correlated with $Y$ but perhaps incorrectly centered or scaled). Nonetheless, the expert predictions clearly contain information (cf. $\text{sign}(U)$) that is not captured by the algorithm.

However, because $U$ is not recorded in the available data, it is not obvious how to distinguish the above scenario from one in which the expert only extracts signal from $X$. For example, they might instead make predictions as follows:

$$\hat{Y} = \text{sign}(X) + \epsilon_2. \tag{3}$$

While a learning algorithm may outperform the expert in both cases, the expert in scenario (2) still captures valuable information; the expert in (3) clearly does not. The goal of this work will be to develop a test which allows us to distinguish between scenarios like these without the strong modeling assumptions made in this example.

---

[1]For example, a well-known study by Eren and Mocan (2018) demonstrates that unexpected losses by the Louisiana State University football team lead judges to hand out longer juvenile sentences; this is a form of capricious decision making which will manifest as noise ($\epsilon_2$) in an analysis of sentencing decisions.

**Contributions.** To understand whether human experts can add value for a given prediction task, we develop a statistical framework under which answering this question becomes a natural hypothesis test. We then provide a simple, data-driven procedure to test this hypothesis. Our proposed algorithm takes the form of a conditional independence test, and is inspired by the Model-X Knockoffs framework of Candès et al. (2016), the Conditional Permutation Test of Berrett et al. (2018) and the 'Model-Powered' test of Sen et al. (2017). Our test is straightforward to implement and provides transparent, interpretable p-values.

Our work is closely related to a large body of literature comparing human performance to that of an algorithm (Cowgill (2018), Dawes et al. (1989), Grove et al. (2000), among others), and developing learning algorithms which are complementary to human expertise (Madras et al. (2018), Raghu et al. (2019), Mozannar and Sontag (2020), Keswani et al. (2021), Agrawal et al. (2018), Bansal et al. (2020), Rastogi et al. (2022) and Bastani et al. (2021)). However, although similarly motivated, we address a different problem which is in a sense 'upstream' of these works, as we are interested in *testing* for whether a human forecaster demonstrates expertise which cannot be replicated by any algorithm. Thus, we think of our test as assessing as a necessary condition for achieving human-AI complementarity; success in practice will further depend on the ability of a mechanism designer to actually incorporate human expertise into some particular algorithmic pipeline or feedback system. We discuss these connections further in Appendix A.

We apply our test to evaluate whether emergency room physicians incorporate valuable information which is not available to a common algorithmic risk score for patients with acute gastrointestinal bleeding (AGIB). To that end, we utilize patient admissions data collected from the emergency department of a large academic hospital system. Consistent with prior literature, we find that this algorithmic score is an exceptionally sensitive measure of patient risk – and one that is highly competitive with physicians' expert assessments. Nonetheless, our test provides strong evidence that physician decisions to either hospitalize or discharge patients with AGIB are incorporating valuable information that is not captured by the screening tool. Our results highlight that prediction accuracy is not sufficient to justify automation of a given prediction task. Instead, our results make a case for experts working *with* a predictive algorithm, even when algorithms might handily outperform their human counterparts.

**Organization.** In Section 2, we formalize the problem of auditing for expertise, and in Section 3 we present our data-driven hypothesis test. Section 4 then examines the theoretical properties of the test. In Section 5 we present our empirical findings from applying this test to real-world patient triage data. Finally, Section 6 provides discussion of our results and directions for future work. We also include a discussion of additional related work in Appendix A, and provide numerical simulations to corroborate our theoretical and empirical results in Appendix F[2].

## 2 Setup and question of interest

We consider a generic prediction task in which the goal is to forecast some outcome $Y \in \mathbb{R}$ on the basis of observable features $X \in \mathcal{X}$. The human expert may additionally have access to some auxiliary private information $U \in \mathcal{U}$. For concreteness, let $\mathcal{X} = \mathbb{R}^d$ for some $d \geq 1$.

We posit that the outcome $Y$ is generated as follows: for some unknown function $f : \mathcal{X} \times \mathcal{U} \to \mathbb{R}$,

$$Y = f(X, U) + \epsilon_1, \tag{4}$$

where, without loss of generality, $\epsilon_1$ represents mean zero idiosyncratic noise with unknown variance.

We are also given predictions by a human expert, denoted as $\hat{Y}$. We posit that the expert predictions $\hat{Y}$ are generated as follows: for some unknown function $\hat{f} : \mathcal{X} \times \mathcal{U} \to \mathbb{R}$,

$$\hat{Y} = \hat{f}(X, U) + \epsilon_2, \tag{5}$$

where $\epsilon_2$ also captures mean zero idiosyncratic noise with unknown variance.

We observe $(X, Y, \hat{Y})$ which obey (4)-(5); the private auxiliary feature $U$ is not observed. Concretely, we observe $n$ data points $(x_i, y_i, \hat{y}_i), i \in [n] \equiv \{1, \ldots, n\}$.

---

[2]Code, data and instructions to replicate our experiments are available at https://github.com/ralur/auditing-human-expertise. Publication of the results and data associated with the empirical study in section 5 have been approved by the relevant institutional review board (IRB).

Our goal is to answer the question "do human experts add information which could not be captured by any algorithm for a given prediction task?" We assume that any competing learning algorithm can utilize $X$ to predict $Y$, but it can not utilize $U$. Thus, our problem reduces to testing whether $U$ has some effect on both $Y$ and $\hat{Y}$, which corresponds to the ability of an expert to extract signal about $Y$ from $U$. If instead $U$ has no effect on $Y$, $\hat{Y}$ or both (either because $U$ is uninformative about $Y$ or the expert is unable to perceive this effect), then conditioned on $X$, $Y$ and $\hat{Y}$ are independent. That is, if the human expert fails to add any information which could not be extracted from the observable features $X$, the following must hold:

$$H_0 : Y \perp\!\!\!\perp \hat{Y} \mid X. \tag{6}$$

Intuitively, $H_0$ captures the fact that once we observe $X$, $\hat{Y}$ provides no *additional* information about $Y$ unless the expert is also making use of some unobserved signal $U$ (whether explicitly or implicitly). In contrast, the rejection of $H_0$ should be taken as an evidence that the expert (or experts) can add value to *any* learning algorithm trained on the observable features $X \in \mathcal{X}$; indeed, a strength of this framework is that it does not require specifying a particular algorithmic baseline. However, it's worth remarking that an important special case is to take $X$ to be the prediction made by some *specific* learning algorithm trained to forecast $Y$. In this setting, our test then reduces to assessing whether $\hat{Y}$ adds information to the predictions made by this learning algorithm, and can be viewed as a form of feature selection.

**Goal.** Test the null hypothesis $H_0$ using observed data $(x_i, y_i, \hat{y}_i), i \in [n] \equiv \{1, \ldots, n\}$.

To make this model concrete, in Section 5 we use our framework to test whether emergency room physicians incorporate information that is not available to a common algorithmic risk score when deciding whether to hospitalize patients. Accordingly, we let $X \in [0, 1]^9$ be the inputs to the risk score, $\hat{Y} \in \{0, 1\}$ be a binary variable indicating whether a given patient was hospitalized, and $Y \in \{0, 1\}$ be an indicator for whether, in retrospect, a patient *should* have been hospitalized. The risk score alone turns out to be a highly accurate predictor of $Y$, but physicians take many other factors into account when making hospitalization decisions. We thus seek to test whether physicians indeed extract *signal* which is not available to the risk score ($Y \not\perp\!\!\!\perp \hat{Y} \mid X$), or whether attempts to incorporate other information and/or exercise expert judgement simply manifest as noise ($Y \perp\!\!\!\perp \hat{Y} \mid X$).

## 3 ExpertTest: a statistical test for human expertise

To derive a statistical test of $H_0$, we will make use of the following elementary but powerful fact about exchangeable random variables.

**A test for exchangeability.** Consider $K + 1$ random variables $(Z_0, \ldots, Z_K)$ which are exchangeable, i.e. the joint distribution of $(Z_0, \ldots, Z_K)$ is identical to that of $(Z_{\sigma(0)}, \ldots, Z_{\sigma(K)})$ for any permutation $\sigma : \{0, \ldots, K\} \to \{0, \ldots, K\}$. For example, if $Z_0, \ldots, Z_K$ are independent and identically distributed (i.i.d.), then they are exchangeable. Let $F$ be a function that maps these variables to a real value. For any such $F(\cdot)$, it can be verified that the order statistics (with any ties broken uniformly at random) of $F(Z_0), \ldots, F(Z_K)$ are uniformly distributed over $(K + 1)!$ permutations of $\{0, \ldots, K\}$. That is, $\tau_K$ defined next, is distributed uniformly over $\{0, 1/K, 2/K, \ldots, 1\}$:

$$\tau_K = \frac{1}{K} \sum_{k=1}^{K} \mathbb{1}[F(Z_0) \lesssim F(Z_k)] \tag{7}$$

where we use definition $\mathbb{1}[\alpha \lesssim \beta] = 1$ if $\alpha < \beta$ and 0 if $\alpha > \beta$. If instead $\alpha = \beta$, we independently assign it to be 1 or 0 with equal probability. Thus, if $(Z_0, \ldots, Z_K)$ are exchangeable, then $\mathbb{P}(\tau_K \leq \alpha) \leq \alpha + 1/(K + 1) \overset{K \to \infty}{\to} \alpha$ and we can reject the hypothesis that $(Z_0, \ldots, Z_K)$ are exchangeable with p-value (effectively) equal to $\tau_K$.

Observe that while this validity guarantee holds for any choice of $F(\cdot)$, the *power* of the test will depend crucially on this choice; for example, a constant function which maps every argument to the same value would have no power to reject the null hypothesis. We return to the choice of $F(\cdot)$ below.

**Constructing exchangeable distributions.** We will leverage the prior fact about the order statistics of exchangeable random variables to design a test of $H_0 : Y \perp\!\!\!\perp \hat{Y} \mid X$. In particular, we would like to use the observed data to construct $K + 1$ random variables that are exchangeable under $H_0$, but not exchangeable otherwise. To that end, consider a simplified setting where $n = 2$, with $x_1 = x_2 = x$. Thus, our observations are $Z_0 = \{(x, y_1, \hat{y}_1), (x, y_2, \hat{y}_2)\}$. Suppose we now sample $(\tilde{y}_1, \tilde{y}_2)$ uniformly at random from $\{(\hat{y}_1, \hat{y}_2), (\hat{y}_2, \hat{y}_1)\}$. That is, we *swap* the observed values $(\hat{y}_1, \hat{y}_2)$ with probability $\frac{1}{2}$ to construct a new dataset $Z_1 = \{(x, y_1, \tilde{y}_1), (x, y_2, \tilde{y}_2)\}$.

Under $H_0$, it is straightforward to show that $Z_0$ and $Z_1$ are independent and identically distributed conditioned on observing $(x, x), (y_1, y_2)$ and either $(\hat{y}_1, \hat{y}_2)$ *or* $(\hat{y}_2, \hat{y}_1)$. That is, $Z_0, Z_1$ are exchangeable, which will allow us to utilize the test described above for $H_0$.

Why condition on this somewhat complicated event? Intuitively, we would like to resample $\tilde{Y} = (\tilde{y}_1, \tilde{y}_2)$ from the distribution of $\mathcal{D}_{\hat{Y}|X}$; under the null, $(x, y, \hat{y})$ and $(x, y, \tilde{y})$ will be exchangeable by definition. However, this requires that we know (or can accurately estimate) the distribution of $\hat{Y} \mid X$, which in turn requires modeling the expert's decision making directly. Instead, we simplify the resampling process by only considering a *swap* of the observed $\hat{y}$ values between identical values of $x$ – this guarantees exchangeable data without modeling $\mathcal{D}_{\hat{Y}|X}$ at all!

This approach can be extended for $n$ larger than 2. Specifically, if there are $L$ pairs of identical $x$ values, i.e. $x_{2\ell-1} = x_{2\ell}$ for $1 \le \ell \le L$, then it is possible to construct i.i.d. $Z_0, \ldots, Z_K$ for larger $K$ by *randomly exchanging* values of $\hat{y}$ for each pair of data points.

As discussed above, we'll also need to choose a particular function $F(\cdot)$ to apply to $Z_0$ and $Z_1$. A natural, discriminatory choice of $F$ is a *loss function*: for example, given $D = \{(x_i, y_i, \hat{y}_i) : i \le 2L\}$, let $F(D) = \sum_i (y_i - \hat{y}_i)^2$. This endows $\tau_K$ with a natural interpretation – it is the probability that an expert could have performed as well as they did (with respect to the chosen loss function $F$) by pure chance, *without* systematically leveraging some unobserved $U$.

Of course, in practice we are unlikely to observe many pairs where $x_{2\ell-1} = x_{2\ell}$, particularly when $x$ takes value in a non-finite domain, e.g. $[0, 1]$ or $\mathbb{R}$. However, if the conditional distribution of $\hat{Y}|X$ is nearly the same for *close enough* values of $X = x$ and $X = x'$, then we can use a similar approach with some additional approximation error. This is precisely the test that we describe next.

**ExpertTest.** Let $L \ge 1$ be an algorithmic parameter and $m : \mathcal{X} \times \mathcal{X} \to \mathbb{R}_{\ge 0}$ be some distance metric over $\mathcal{X}$, e.g. the $\ell_2$ distance. Let $F(\cdot)$ be some loss function of interest, e.g. the mean squared error.

First, compute $m(x_i, x_j) : i \ne j \in [n]$ and greedily select $L$ disjoint pairs which are as close as possible under $m(\cdot, \cdot)$. Denote these pairs by $\{(x_{i_{2\ell-1}}, x_{i_{2\ell}}) : \ell \in [L]\}$.

Let $D_0 = \{(x_i, y_i, \hat{y}_i) : i \in \{i_{2\ell-1}, i_{2\ell} : \ell \in [L]\}\}$ denote the observed dataset restricted to the $L$ chosen pairs. Let $D_1$ be an additional dataset generated by independently swapping each pair $(\hat{y}_{i_{2\ell-1}}, \hat{y}_{i_{2\ell}})$ with probability $1/2$, and repeat this resampling procedure to generate $D_1 \ldots D_K$. Next, compute $\tau_K$ as follows:

$$\tau_K = \frac{1}{K} \sum_{k=1}^{K} \mathbb{1}[F(D_0) \lesssim F(D_k)] \tag{8}$$

Finally, we reject the hypothesis $H_0$ with $p$-value $\alpha + 1/(K+1)$ if $\tau_K \le \alpha$ for any desired confidence level $\alpha \in (0, 1)$. Our test is thus quite simple: find $L$ pairs of points that are close under some distance metric $m(\cdot, \cdot)$, and create $K$ synthetic datasets by swapping the expert forecasts for each pair independently with probability $1/2$. If the expert's loss on the original dataset is "small" relative to the loss on these resampled datasets, this is evidence that the synthetic datasets are not exchangeable with the original, and thus, the expert is using some private information $U$.

Of course, unlike in the example above, we swapped pairs of predictions for *different* values of $x$. Thus, $D_0 \ldots D_K$ are not exchangeable under $H_0$. However, we'll argue that because we paired "nearby" values of $x$, these datasets are "nearly" exchangeable. These are the results we present next.

## 4 Results

We provide theoretical guarantees associated with the **ExpertTest**. First, we demonstrate the validity of our test in a generic setting. That is, if $H_0$ is true, then **ExpertTest** will not reject it with high probability. We then quantify this guarantee precisely under a meaningful generative model.

To state the validity result, we need some notation. For any $(x, \hat{y})$ and $(x', \hat{y}')$, define the odds ratio

$$r((x, \hat{y}), (x', \hat{y}')) = \frac{Q(\hat{Y} = \hat{y} \mid X = x) \times Q(\hat{Y} = \hat{y}' \mid X = x')}{Q(\hat{Y} = \hat{y}' \mid X = x) \times Q(\hat{Y} = \hat{y} \mid X = x')}, \tag{9}$$

where $Q(\cdot | \cdot)$ represents the density of the conditional distribution of human predictions $\hat{Y}|X$ under $H_0$. For simplicity, we assume that such a conditional density exists.

**Theorem 1 (Validity of ExpertTest)** *Given $\alpha \in (0, 1)$ and parameters $K \geq 1, L \geq 1$, the Type I error of **ExpertTest** satisfies*

$$\mathbb{P}(\tau_K \leq \alpha) \leq \alpha + \left(1 - (1 - \varepsilon_{n,L}^*)^L\right) + \frac{1}{K+1}. \tag{10}$$

*Where $\varepsilon_{n,L}^*$ is defined as follows*

$$\varepsilon_{n,L}^* = \max_{\ell \in [L]} \left| \frac{1}{1 + r((x_{i_{2\ell-1}}, \hat{y}_{i_{2\ell-1}}), (x_{i_{2\ell}}, \hat{y}_{i_{2\ell}}))} - \frac{1}{2} \right|. \tag{11}$$

We remark briefly on the role of the parameters $L$ and $K$ in this result. To begin with, $\frac{1}{K+1}$ is embedded in the type I error, and thus taking the number of resampled datasets $K$ to be as large as possible (subject only to computational constraints) sharpens the validity guarantee. We also observe that the bound becomes weaker as $L$ increases. However, observe also that **ExpertTest** is implicitly using an $L$-dimensional distribution (or $L$ fresh samples) to reject $H_0$, which means that increasing $L$ also provides additional *power* to the test.

Notice also that the odds ratio (9) is guaranteed to be 1 if $x = x'$, regardless of the underlying distribution $\mathcal{D}$. This is not a coincidence, and our test is based implicitly on the heuristic that the odds ratio will tend away from 1 as the distance $m(x, x')$ increases (we quantify this intuition precisely below). Thus, increasing $L$ will typically also increase $\varepsilon_{n,L}^*$, because larger values of $L$ will force us to pair additional observations $(x, x')$ which are farther apart under the distance metric.

The type one error bound (10) suggests that we can balance the trade off between validity and power when $\varepsilon_{n,L}^* L \ll 1$ or $o(1)$, as the right hand side of (10) reduces to $\alpha + o(1)$. Next we describe a representative generative setup where there is a natural choice of $L$ that leads to $\varepsilon_{n,L}^* L = o(1)$.

**Generative model.** Let $\mathcal{X} = [0, 1]^d \subset \mathbb{R}^d$. Let the conditional density of the human expert's forecasts $Q(\cdot|x)$ be *smooth*. Specifically, for any $x, x' \in [0, 1]^d$,

$$\sup_{\hat{y} \in \mathbb{R}} \frac{Q(\hat{y} \mid X = x)}{Q(\hat{y} \mid X = x')} \leq 1 + C \times \|x - x'\|_2, \tag{12}$$

for some constant $C > 0$. Under this setup, Theorem 1 reduces to the following.

**Theorem 2 (Asymptotic Validity)** *Given $\alpha \in (0, 1)$ and under (12), with the appropriate choice of $L \geq 1$, the type I error of **ExpertTest** satisfies*

$$\mathbb{P}(\tau_K \leq \alpha) \leq \alpha + o(1). \tag{13}$$

*as $n, K \to \infty$.*

Intuitively, (12) is intended to model a forecasting rule which is 'simple,' in the sense that human experts don't finely distinguish between instances whose feature vectors are close under the $\ell_2$ norm. Importantly, this does not rule out the possibility that predictions for two specific $(x, x')$ instances could differ substantially – only that the distributions $\hat{Y} \mid X = x$ and $\hat{Y} \mid X = x'$ are similar when $x \approx x'$. We make no such assumption about $\mathcal{D}_{Y|X}$, the conditional distribution of the true outcomes.

Proofs of theorems 1 and 2 can be found in Appendices B and C respectively. We now illustrate the utility of our test with an empirical study of physician hospitalization decisions.

# 5 A case study: physician expertise in emergency room triage

Emergency room triage decisions present a natural real-world setting for our work, as we can assess whether physicians make hospitalization decisions by incorporating information which is not available to an algorithmic risk score. We consider the particular case of patients who present in the emergency room with acute gastrointestinal bleeding (hereafter referred to as AGIB), and assess whether physicians' decisions to either hospitalize or discharge each patient appear to be capturing information which is not available to the Glasgow-Blatchford Score (GBS). The GBS is a standardized measure of risk which is known to be a highly sensitive indicator for whether a patient presenting with AGIB will indeed require hospitalization (findings which we corroborate below). However, despite the excellent performance of this algorithmic risk score, we might be understandably hesitant to automate triage decisions without any physician oversight. As just one example, anticoagulant medications ('blood thinners') are known to exacerbate the risk of severe bleeding. However, whether or not a patient is taking anticoagulant medication is not included as a feature in the construction of the Glasgow-Blatchford score, and indeed may not even be recorded in the patient's electronic health record (if, for example, they are a member of an underserved population and have had limited prior contact with the healthcare system). This is one of many additional factors an emergency room physician might elicit directly from the patient to inform an admit/discharge decision. We thus seek to answer the following question:

*Do emergency room physicians usefully incorporate information which is not available to the Glasgow-Blatchford score?*

We answer in the affirmative, demonstrating that although the GBS provides risk scores which are highly competitive with (and indeed, arguably better than) physicians' discretionary decisions, there is strong evidence that physicians are incorporating additional information which is not captured in the construction of the GBS. Before presenting our results, we first provide additional background about this setting.

**Background: risk stratification and triage for gastrointestinal bleeding.** Acute gastrointestinal bleeding is a potentially serious condition for which 530,855 patients/year receive treatment in the United States alone (Peery et al. (2022)). It is estimated that 32% of patients with presumed bleeding from the lower gastrointestinal tract (Oakland et al. (2017)) and 45% of patients with presumed bleeding from the upper gastrointestinal tract (Stanley et al. (2017)) require urgent medical intervention; overall mortality rates for AGIB in the U.S. are estimated at around 3 per $100,000$ (Peery et al. (2022)). For patients who present with AGIB in the emergency room, the attending physician is tasked with deciding whether the bleeding is severe enough to warrant admission to the hospital. However, the specific etiology of AGIB is often difficult to determine from patient presentation alone, and gold standard diagnostic techniques – an endoscopy for upper GI bleeding or a colonoscopy for lower GI bleeding – are both invasive and costly, particularly when performed urgently in a hospital setting. To aid emergency room physicians in making this determination more efficiently, the Glasgow-Blatchford Bleeding Score or GBS (Blatchford et al. (2000)) is a standard screening metric used to assess the risk that a patient with acute upper GI bleeding will require red blood cell transfusion, intervention to stop bleeding, or die within 30 days. It has been also validated in patients with acute lower gastrointestinal bleeding[3] to assess need for intervention to stop bleeding or risk of death (Asad Ur-Rahman and Abusaada (2018)); accordingly, we interpret the GBS as a measure of risk for patients who present with either upper or lower GI bleeding in the emergency department.

**Construction of the Glasgow-Blatchford Score.** The Glasgow-Blatchford Score is a function of the following nine patient characteristics: blood urea nitrogen (BUN), hemoglobin (HGB), systolic blood pressure (SBP), pulse, cardiac failure, hepatic disease, melena, syncope and biological sex. The first four are real-valued and the latter five are binary. The GBS is calculated by first converting the continuous features to ordinal values (BUN and HGB to 6 point scales, SBP to a 3 point scale and pulse to a binary value) and then summing the values of the first 8 features. Biological sex is used to inform the conversion of HGB to an ordinal value. Scores are integers ranging from 0 to 23,

---

[3]US and international guidelines use the Glasgow-Blatchford score as the preferred risk score for assessing patients with upper gastrointestinal bleeding (Laine et al. (2021); Barkun et al. (2019)). Other risk scores tailored to bleeding in the lower gastrointestinal tract have been proposed in the literature, but these are less commonly used in practice. We refer interested readers to Almaghrabi et al. (2022) for additional details.

with higher scores indicating a higher risk that a patient will require subsequent intervention. US and international guidelines suggest that patients with a score of 0 or 1 can be safely discharged from the emergency department (Laine et al. (2021); Barkun et al. (2019)), with further investigation to be performed outside the hospital. For additional details on the construction of the GBS, we refer to Blatchford et al. (2000).

**Defining features, predictions and outcomes.** We consider a sample of 3617 patients who presented with AGIB at one of three hospitals in a large academic health system between 2014 and 2018. Consistent with the goals of triage for patients with AGIB, we record an 'adverse outcome' if a patient (1) requires some form of urgent intervention to stop bleeding (endoscopic, interventional radiologic, or surgical; excluding patients who only undergo a diagnostic endoscopy or colonoscopy) while in the hospital (2) dies within 30 days of their emergency room visit or (3) is initially discharged but later readmitted within 30 days.[4] As is typical of large urban hospitals in the United States, staffing protocols at this health system dictate a separation of responsibilities between emergency room physicians and other specialists. In particular, while emergency room physicians make an initial decision whether to hospitalize a patient, it is typically a gastrointestinal specialist who subsequently decides whether a patient admitted with AGIB requires some form of urgent hemostatic intervention. This feature, along with our ability to observe whether discharged patients are subsequently readmitted, substantially mitigates the selective labels issue that might otherwise occur in this setting. Thus, consistent with clinical and regulatory guidelines – to avoid hospitalizing patients who do not require urgent intervention (Stanley et al. (2009)), and to avoid discharging patients who are likely to be readmitted within 30 days (NEJM (2018)) – we interpret the emergency room physician's decision to admit or discharge a patient as a *prediction* that one of these adverse outcomes will occur. We thus instantiate our model by letting $X_i \in [0,1]^9$ be the nine discrete patient characteristics from which the Glasgow-Blatchford Score is computed[5]; the only transformation we apply is to normalize each feature to lie in $[0,1]$. We further let $\hat{Y}_i \in \{0,1\}$ indicate whether that patient was initially hospitalized, and $Y_i \in \{0,1\}$ indicate whether that patient suffered one of the adverse outcomes defined above.

**Assessing the accuracy of physician decisions.** We first summarize the performance of the emergency room physicians' hospitalization decisions, and compare them with the performance of a simple rule which would instead admit every patient with a GBS above a certain threshold and discharge the remainder (Table 1). We consider thresholds of 0 and 1 – the generally accepted range for low risk patients (Laine et al. (2021); Barkun et al. (2019)) – as well as less conservative thresholds of 2 and 7 (the latter of which we find maximizes overall accuracy). For additional context, we also provide the total fraction of patients admitted under each decision rule.

| Decision Rule | Fraction Hospitalized | Accuracy | Sensitivity | Specificity |
|---|---|---|---|---|
| Physician Discretion | $0.86 \pm 0.02$ | $0.55 \pm 0.02$ | $0.99 \pm 0.00$ | $0.24 \pm 0.02$ |
| Admit GBS > 0 | $0.88 \pm 0.02$ | $0.53 \pm 0.02$ | $0.99 \pm 0.00$ | $0.19 \pm 0.02$ |
| Admit GBS > 1 | $0.80 \pm 0.02$ | $0.60 \pm 0.02$ | $0.98 \pm 0.00$ | $0.33 \pm 0.02$ |
| Admit GBS > 2 | $0.73 \pm 0.02$ | $0.66 \pm 0.02$ | $0.97 \pm 0.00$ | $0.43 \pm 0.02$ |
| Admit GBS > 7 | $0.40 \pm 0.02$ | $0.79 \pm 0.02$ | $0.73 \pm 0.02$ | $0.84 \pm 0.02$ |

Table 1: Comparing the accuracy of physician hospitalization decisions ('Physician Discretion') to those made by thresholding the GBS. For example, 'Admit GBS > 1' hospitalizes patients with a GBS strictly larger than 1. 'Fraction Hospitalized' indicates the fraction of patients hospitalized by each rule. 'Accuracy' indicates the 0/1 accuracy of each rule, where a decision is correct if it hospitalizes a patient who suffers an adverse outcome (as defined above) or discharges a patient who does not. 'Sensitivity' indicates the fraction of patients who suffer an adverse outcome that are correctly hospitalized, and 'Specificity' indicates the fraction of patients who do not suffer an adverse outcome that are correctly discharged. Results are reported to $\pm 2$ standard errors.

---

[4]This threshold is consistent with the definition used in the Centers for Medicare and Medicaid Services Hospital Readmission Reduction Program, which seeks to incentivize healthcare providers to avoid discharging patients who will be readmitted within 30 days (NEJM (2018))

[5]We consider alternative definitions of the feature space in Appendix E, where we find substantively similar results.

Unsurprisingly, we find that the physicians calibrate their decisions to maximize sensitivity (minimize false negatives) at the expense of admitting a significant fraction of patients who, in retrospect, could have been discharged immediately. Indeed we find that although $86\%$ of patients are initially hospitalized, only $\approx 42\%$ actually require a specific hospital-based intervention or otherwise suffer an adverse outcome which would justify hospitalization (i.e., $y_i = 1$). Consistent with Blatchford et al. (2000) and Chatten et al. (2018), we also find that thresholding the GBS in the range of $[0, 2]$ achieves a sensitivity of close to $100\%$. We further can see that using one of these thresholds may achieve overall accuracy (driven by improved specificity) which is substantially better than physician discretion. Nonetheless, we seek to test whether physicians demonstrate evidence of expertise in distinguishing patients with identical (or nearly identical) scores.

**Testing for physician expertise.** We now present the results of running **ExpertTest** for $K = 1000$ resampled datasets and various values of $L$ (where $L = 1808$ is the largest possible choice given $n = 3617$) in Table 2. We define the distance metric $m(x_1, x_2) := \|x_1 - x_2\|_2$, though this choice is inconsequential when the number of 'mismatched pairs' (those pairs $x, x'$ where $x \neq x'$) is 0.

We also observe that, in the special case of binary predictions and outcomes, it is possible to analytically determine the number of swaps which increase or decrease the value of nearly any natural loss function $F(\cdot)$. Thus, although we let $F(D) := \frac{1}{n} \sum_i \mathbb{1}[y_i \neq \hat{y}_i]$ for concreteness, our results are largely insensitive to this choice; in particular, they remain the same when false negatives and false positives might incur arbitrarily different costs. We elaborate on this phenomenon in Appendix E.

| L | mismatched pairs | swaps that increase loss | swaps that decrease loss | $\tau$ |
|---|---|---|---|---|
| 100 | 0 | 5 | 1 | 0.061 |
| 250 | 0 | 12 | 1 | 0.003 |
| 500 | 0 | 21 | 2 | <.001 |
| 1000 | 0 | 42 | 2 | <.001 |
| 1808 | 265 | 66 | 4 | <.001 |

Table 2: The results of running ExpertTest, where each pair of patients is chosen to be as similar as possible with respect to the nine (discrete) inputs to the Glasgow-Blatchford score. $L$ indicates the number of pairs selected for the test, of which 'mismatched pairs' are not identical to each other. Swaps that decrease (respectively, increase) loss indicates how many of the $L$ pairs result in a decrease (respectively, increase) in the 0/1 loss when their corresponding hospitalization decisions are exchanged with each other. $\tau$ is the p-value obtained from running ExpertTest.

As the results demonstrate, there is *very strong evidence that emergency room physicians incorporate information which is not available to the Glasgow-Blatchford score*. In particular, our test indicates that physicians can reliably distinguish patients who appear identical with respect to the nine features considered by the GBS – and can make hospitalization decisions accordingly – even though simple GBS thresholding is highly competitive with physician performance. This implies that *no* predictive algorithm trained on these nine features, even one which is substantially more complicated than the GBS, can fully capture the information that physicians use to make hospitalization decisions.

To interpret the value of $\tau$, observe that for $L \geq 500$ we recover the smallest possible value $\tau = 1/(K + 1) = 1/1001$. Furthermore, for all but the final experiment, the number of mismatched pairs is 0, which means there is no additional type one error incurred (i.e., $\varepsilon_{n,L}^*$ (11) is guaranteed to be 0). For additional intuition on the behavior of **ExpertTest**, we refer the reader to the synthetic experiments in Appendix F.

## 6 Discussion and limitations

In this work we provide a simple test to detect whether a human forecaster is incorporating unobserved information into their predictions, and illustrate its utility in a case study of hospitalization decisions made by emergency room physicians. A key insight is to recognize that this requires more care than simply testing whether the forecaster *outperforms* an algorithm trained on observable data; indeed, a large body of prior work suggests that this is rarely the case. Nonetheless, there are many settings in which we might expect that an expert is using information or intuition which is difficult to replicate with a predictive model.

An important limitation of our approach is that we do not consider the possibility that expert forecasts might inform decisions which causally effect the outcome of interest, as is often the case in practice. We also do not address the possibility that the objective of interest is not merely accuracy, but perhaps some more sophisticated measure of utility (e.g., one which also values fairness or simplicity); this is explored in Rambachan (2022). We caution more generally that there are often normative reasons to prefer human decision makers, and our test captures merely one possible notion of expertise. The results of our test should thus not be taken as *recommending* the automation of a given forecasting task.

Furthermore, while our framework is quite general, it is worth emphasizing that the specific algorithm we propose is only one possible test of $H_0$. Our algorithm does not scale naturally to settings where $\mathcal{X}$ is high-dimensional, and in such cases it is likely that a more sophisticated test of conditional independence (e.g. a kernel-based method; see Fukumizu et al. (2004), Gretton et al. (2007) and Zhang et al. (2011), among others) would have more power to reject $H_0$. Another possible heuristic is to simply choose some learning algorithm to estimate (e.g.) $\mathbb{E}[Y \mid X]$ and $\mathbb{E}[Y \mid X, \hat{Y}]$, and examine which of the two provides better out of sample performance. This can be viewed as a form of feature selection; indeed the 'knockoffs' approach of Candès et al. (2016) which inspires our work is often used as a feature selection procedure in machine learning pipelines. However, most learning algorithms do not provide p-values with the same natural interpretation we describe in section 3, and we thus view these approaches as complementary to our own.

Finally, our work draws a clean separation between the 'upstream' inferential goal of detecting whether a forecaster is incorporating unobserved information and the 'downstream' algorithmic task of designing tools which complement or otherwise incorporate human expertise. These problems share a very similar underlying structure however, and we conjecture that – as has been observed in other supervised learning settings, e.g. Kearns et al. (2018) – there is a tight connection between these auditing and learning problems. We leave an exploration of these questions for future work.

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

# A    Additional related work

**Human decision making, interplay with algorithms.** Our work contributes to a vast literature on understanding how humans, and particularly human experts, make decisions. We do not attempt to provide a comprehensive summary of this work, but refer the reader to Tversky and Kahneman (1974) and Camerer and Johnson (1991) for general background. Of particular relevance for our setting is work which investigates whether humans make *systematic* mistakes in their decisions, which has been studied in the context of bail decisions (Kleinberg et al. (2017), Rambachan (2022), Lakkaraju et al. (2017) and Arnold et al. (2020)), college admissions (Kuncel et al. (2013), Dawes (1971)) and patient triage and diagnosis (Currie and MacLeod (2017), Mullainathan and Obermeyer (2019)) among others. One common theme in these works is that the decision made by the human expert will often influence the outcome of interest; for example, an emergency room doctor's initial diagnosis will inform the treatment a patient receives, which subsequently affects their health outcomes. Furthermore, it is often the case that even observing the outcome of interest is contingent on the human's decision: for example, in a college admissions setting, we might only observe historical outcomes for *admitted* students, which makes it challenging to draw inferences about *applicants*. This one-sided labeling problem is a form of endogeneity which has been well studied in the context of causal inference, and these works often adopt a causal perspective to address these challenges.

As discussed in Section 6, our instead work assumes that all outcomes are observable and, importantly, that they are not affected by the human predictions. We also do not explicitly grapple with whether the human expert has an objective other than maximizing accuracy under a known metric (e.g., squared error). Though this is often a primary concern in many high-stakes settings – for example, ensuring that bail decisions are not only accurate but also nondiscriminatory – it is outside the scope of our work, and we refer the reader instead to Rambachan (2022) for further discussion.

As discussed in section 1, another closely related theme is directly comparing human performance to that of an algorithm (Cowgill (2018), Dawes et al. (1989), Grove et al. (2000)), and developing learning algorithms which are complementary to human expertise (Madras et al. (2018), Raghu et al. (2019), Mozannar and Sontag (2020), Keswani et al. (2021), Agrawal et al. (2018) and Bastani et al. (2021)). A key design consideration when designing algorithms to complement human expertise involves reasoning about the ways in which humans may *respond* to the introduction of an algorithm, which may be strategic (e.g. Kleinberg and Raghavan (2018), Perdomo et al. (2020), Cen and Shah (2021), Hardt et al. (2015), Liu et al. (2020)) or subject to behavioral biases (Kleinberg et al. (2022)). These behaviors can make it challenging to design algorithms which work with humans to achieve the desired outcomes, as humans may respond to algorithmic recommendations or feedback in unpredictable ways.

**Conditional independence testing.** We cast our setting as a special case of conditional independence testing, which has been well studied in the statistics community. For background we refer the reader to Dawid (1979). It has long been known that testing conditional independence between three (possibly high-dimensional) random variables is a challenging problem, and the recent result of Shah and Peters (2018) demonstrates that this is in fact impossible in full generality. Nonetheless, there are many methods for testing conditional independence under natural assumptions; perhaps the most popular are the kernel-based methods introduced by Fukumizu et al. (2004) and subsequently developed in Gretton et al. (2007) and Zhang et al. (2011), among others.

Our work instead takes inspiration from the 'knockoffs' framework developed in Candès et al. (2016), Barber et al. (2018) and Barber and Candès (2019), as well as the closely related conditional permutation test of Berrett et al. (2018). These works leverage the elementary observation that, under the null hypothesis that (specialized to our notation) the outcome $Y$ and prediction $\hat{Y}$ are independent conditional on the observed data $X$, new samples from the distribution of $\hat{Y} \mid X$ should be exchangeable with $\hat{Y}$. Thus, if we know – or can accurately estimate – the distribution of $\hat{Y} \mid X$, it is straightforward to generate fresh samples ('knockoffs') which are statistically indistinguishable from the original data under the null hypothesis $H_0 : Y \perp\!\!\!\perp \hat{Y} \mid X$. Thus, if the observed data appears anomalous with respect to these knockoffs, this may provide us a basis on which to reject $H_0$.

Our work avoids takes inspiration from this framework, but avoids estimating the distribution of $\hat{Y} \mid X$ by instead leveraging a simple nearest-neighbors style algorithm for generating knockoffs. In that sense, our technique builds upon the nearest-neighbors based estimator of Runge (2017), and is

nearly identical to the one-nearest-neighbor procedure proposed in the 'model-powered' conditional independence test of Sen et al. (2017). This algorithm is a subroutine in their more complicated end-to-end procedure, which involves training a model to distinguish between the observed data and knockoffs generated via swapping the 'predictions' (again specializing their general test to our setting) associated with instances which are as close as possible under the $\ell_2$ norm. By contrast, we analyze a similar procedure under different smoothness assumptions which allow us to recover p-values that are entirely model free.

## B   Proof of Theorem 1

We establish the proof of Theorem 1 following the intuition presented in Section 3. Specifically, we first bound the type I error of **ExpertTest** in the idealized case where the data set contains $L$ identical pairs of observations $x = x'$. We then refine this bound to handle the case, which is more likely in practice, that the pairs chosen are merely close together. Our final bound thus includes additional approximation error to account for the 'similarity' of the pairs – if we succeed in finding $L$ pairs which are identical, we get nearly exact type I error control, whereas if we are forced to pair instances which are 'far apart', we incur additional approximation error. We formalize this intuition below.

**An idealized bound.** We first establish that $\mathbb{P}(\tau_K \leq \alpha) \leq \alpha + \frac{1}{K+1}$ for any $\alpha \in [0, 1]$ when $x = x'$ for every $(x, x')$ pair chosen by **ExpertTest**.

To that end, we observe $n$ data points $(x_i, y_i, \hat{y}_i)$, $i \in [n]$. Let $\mathcal{L} = \{i_{2\ell-1}, i_{2\ell} : \ell \in [L]\}$ denote the indices of the pairs chosen by **ExpertTest**, with $(x_{i_{2\ell-1}}, x_{i_{2\ell}})$ for $\ell \in [L]$ denoting the pairs themselves.

By assumption, **ExpertTest** succeeds in finding identical pairs:
$$x_{i_{2\ell-1}} = x_{i_{2\ell}}, \ \forall \ \ell \in [L]. \tag{14}$$
Therefore, from the definition (9) it follows that $r((x_{i_{2\ell-1}}, \hat{y}_{i_{2\ell-1}}), (x_{i_{2\ell}}, \hat{y}_{i_{2\ell}})) = 1$ for all $\ell \in [L]$.

As discussed in Section 3, **ExpertTest** will repeatedly generate $n$ fresh data points, denoted by $\tilde{D}$, as follows. For each index $i \in [n] \backslash \mathcal{L}$, i.e. those not corresponding to those selected in $L$ pairs, we select exactly the observed data $(x_i, y_i, \hat{y}_i)$.

For $i \in \mathcal{L}$, we sample a data triplet as follows: for $i \in \{i_{2\ell-1}, i_{2\ell}\}$, we let $(x_{i2\ell-1}, y_{i2\ell-1}), (x_{i2\ell}, y_{i2\ell})$ be the observed values but sample the corresponding $\hat{y}$ values from $\{(\hat{y}_{2\ell-1}, \hat{y}_{2\ell}), (\hat{y}_{2\ell}, \hat{y}_{2\ell-1})\}$ with equal probability. That is, we *swap* the $\hat{y}$ values associated with $(x_{i2\ell-1}, y_{i2\ell-1}), (x_{i2\ell}, y_{i2\ell})$ with probability $\frac{1}{2}$. We argue that this resampling process is implicitly generating a fresh, identically distributed dataset from the underlying distribution $\mathcal{D}$ conditioned on the following event $\mathcal{F}$:

$$\mathcal{F} = \{(x_i, y_i, \hat{y}_i) : i \in [n] \backslash \mathcal{L}\} \cup \{(x_i, y_i) : i \in \mathcal{L}\} \cup \{(\hat{y}_{i_{2\ell-1}}, \hat{y}_{i_{2\ell}}) \vee (\hat{y}_{i_{2\ell}}, \hat{y}_{i_{2\ell-1}}) : \ell \in [L]\}. \tag{15}$$

Why condition on $\mathcal{F}$? As discussed in section 3, a straightforward test would involve simply resampling $K$ fresh datasets from the underlying distribution $\mathcal{D}_X \times \mathcal{D}_{\hat{Y}|X} \times \mathcal{D}_{Y|X}$ and observing that, by definition, these datasets are distributed identically to the observed data $D_0$ under $H_0 : Y \perp\!\!\!\perp \hat{Y} \mid X$. While this would form the basis for a valid test along the lines of the one described in Section 3, it requires knowledge of the underlying distribution which we are unlikely to have in practice. Thus, we instead condition on nearly everything in the observed data – the values and exact ordering of $X$ and the values and exact ordering of $Y$, and the values of $\hat{Y}$ up to a specific set of allowed permutations (those induced by swapping 0 or more paired $\hat{y}_{i_{2\ell-1}}, \hat{y}_{i_{2\ell}}$ values). This substantially simplifies the resampling problem, as we only need to reason about the correct 'swap' probability for each such pair. This can be viewed as an alternative factorization of the underlying distribution $\mathcal{D}$ under $H_0$ – rather than sampling $X \sim \mathcal{D}_X, Y \sim \mathcal{D}_{Y|X}, \hat{Y} \sim \mathcal{D}_{\hat{Y}|X}$, instead sample an event $\mathcal{F} \sim \mathcal{D}_\mathcal{F}$ from the induced distribution over events of the form (15), and then sample $\hat{Y} \sim \mathcal{D}_{\hat{Y}|\mathcal{F}}$.

First, we show that conditional on $\mathcal{F}$, the resampled dataset $\tilde{D}$ and the observed dataset $D_0$ are indeed identically distributed under $H_0 : Y \perp\!\!\!\perp \hat{Y} \mid X$ (that they are also independent, conditional on $\mathcal{F}$, is

clear by construction). To see this, observe that for each $\ell \in [L]$:

$$\mathbb{P}((x_{i_{2\ell-1}}, y_{i_{2\ell-1}}, \hat{y}_{i_{2\ell-1}}), (x_{i_{2\ell}}, y_{i_{2\ell}}, \hat{y}_{i_{2\ell}})) \tag{16}$$

$$= \mathbb{P}(x_{i_{2\ell-1}})\mathbb{P}(y_{i_{2\ell-1}} \mid x_{i_{2\ell-1}})\mathbb{P}(\hat{y}_{i_{2\ell-1}} \mid x_{i_{2\ell-1}})\mathbb{P}(x_{i_{2\ell}})\mathbb{P}(y_{i_{2\ell}} \mid x_{i_{2\ell}})\mathbb{P}(\hat{y}_{i_{2\ell}} \mid x_{i_{2\ell}}) \tag{17}$$

$$= \mathbb{P}(x_{i_{2\ell-1}})\mathbb{P}(y_{i_{2\ell-1}} \mid x_{i_{2\ell-1}})\mathbb{P}(\hat{y}_{i_{2\ell}} \mid x_{i_{2\ell-1}})\mathbb{P}(x_{i_{2\ell}})\mathbb{P}(y_{i_{2\ell}} \mid x_{i_{2\ell}})\mathbb{P}(\hat{y}_{i_{2\ell-1}} \mid x_{i_{2\ell}}) \tag{18}$$

$$= \mathbb{P}((x_{i_{2\ell-1}}, y_{i_{2\ell-1}}, \hat{y}_{i_{2\ell}}), (x_{i_{2\ell}}, y_{i_{2\ell}}, \hat{y}_{i_{2\ell-1}})) \tag{19}$$

In above, (17) follows from $H_0$ and the assumption that the data are drawn i.i.d., and (18) follows from assumption (14) that $x_{i_{2\ell-1}} = x_{i_{2\ell}}$. By construction, the events in (16) and (19) are the only two possible outcomes after conditioning on $\mathcal{F}$, and this simple argument shows that in fact they are equally likely.

Thus, let $\tilde{D}_1, \ldots, \tilde{D}_K$ be $K$ independent and identically distributed datasets generated by the above procedure. Let $\tilde{D}_0$ be one additional sample from this distribution, which we showed was distributed identically to $D_0$ under the idealized assumption (14).

As discussed in Section 3, for any real-valued function $F$ that maps each dataset to $\mathbb{R}$, we have

$$\tau_K = \frac{1}{K} \sum_{k=1}^{K} \mathbb{1}[F(\tilde{D}_0) \lesssim F(\tilde{D}_k)] \tag{20}$$

where we use definition of $\mathbb{1}[\cdot \lesssim \cdot]$ as in (7).

Because $\tilde{D}_0, \ldots, \tilde{D}_K$ are i.i.d., and thus exchangeable, it follows that $\frac{1}{K} \sum_{k=1}^{K} \mathbb{1}[F(\tilde{D}_0) \lesssim F(\tilde{D}_k)]$ is uniformly distributed $\{0, \frac{1}{K}, \ldots, 1\}$. Therefore, with a little algebra it can be verified that for any $\alpha \in [0, 1]$, $\tau_K$ satisfies

$$\mathbb{P}_{\tilde{D}_0, \ldots, \tilde{D}_K \mid \mathcal{F}}(\tau_K \leq \alpha) \leq \alpha + \frac{1}{K+1}. \tag{21}$$

Because $D_0$ and $\tilde{D}_0$ are independent and identically distributed under (14), the same holds if we replace $\tilde{D}_0$ with $D_0$. Thus, **ExpertTest** provides nearly exact type I error control in the case that the idealized assumption (14) holds. This result will serve as a useful building block, as we'll now proceed to relax this assumption and bound the type I error of **ExpertTest** in terms of the total variation distance between $\tilde{D}_0$ and $D_0$.

**Fixing the approximation.** $\tilde{D}_1, \ldots, \tilde{D}_K$ are synthetically generated datasets that are independent and identically distributed. The argument above replaced the observed dataset $D_0$ with a resampled 'idealized' dataset $\tilde{D}_0$, which is also independent and identically distributed with respect to $\tilde{D}_1, \ldots, \tilde{D}_K$, and then used this fact to demonstrate that $\mathbb{P}_{\tilde{D}_0, \ldots, \tilde{D}_K \mid \mathcal{F}}(\tau \leq \alpha) \leq \alpha + \frac{1}{K+1}$. If the idealized assumption (14) holds, replacing $D_0$ with $\tilde{D}_0$ is immaterial as we showed the two are identically distributed conditional on $\mathcal{F}$. Of course, this assumption will not hold in general, and this is what we seek to correct next.

Let $\bar{D}_0 \sim \mathcal{D}_{\cdot \mid \mathcal{F}}$ be a random variable distributed according to the true underlying distribution $\mathcal{D}$, conditional on the event $\mathcal{F}$. The observed data $D_0$ can be interpreted as one realization of this random variable. One way to quantify the excess type I error incurred by using $\tilde{D}_0$ in place of $D_0$ is to bound the total variation distance between the joint distributions of $(\tilde{D}_0, \ldots \tilde{D}_K)$ and that of $(\bar{D}_0, \tilde{D}_1, \ldots \tilde{D}_K)$. Specifically, it follows from the definition of total variation distance that:

$$\mathbb{P}_{\bar{D}_0, \ldots, \tilde{D}_K \mid \mathcal{F}}(\tau_K \leq \alpha) \leq \mathbb{P}_{\tilde{D}_0, \ldots, \tilde{D}_K \mid \mathcal{F}}(\tau_K \leq \alpha) + \mathsf{TV}(\mathbb{P}_{\bar{D}_0, \ldots, \tilde{D}_K \mid \mathcal{F}}, \mathbb{P}_{\tilde{D}_0, \ldots, \tilde{D}_K \mid \mathcal{F}}), \tag{22}$$

where $\mathsf{TV}(\mathbb{P}_{\bar{D}_0, \ldots, \tilde{D}_K \mid \mathcal{F}}, \mathbb{P}_{\tilde{D}_0, \ldots, \tilde{D}_K \mid \mathcal{F}})$ denotes the total variation distance between its arguments. Due to the independence of the resampled datasets, this simplifies to:

$$\mathsf{TV}(\mathbb{P}_{\bar{D}_0, \ldots, \tilde{D}_K \mid \mathcal{F}}, \mathbb{P}_{\tilde{D}_0, \ldots, \tilde{D}_K \mid \mathcal{F}}) = \mathsf{TV}(\mathbb{P}_{\bar{D}_0 \mid \mathcal{F}}, \mathbb{P}_{\tilde{D}_0 \mid \mathcal{F}}). \tag{23}$$

Therefore, we need only bound the total variation distance between $\mathbb{P}_{\bar{D}_0|\mathcal{F}}$ and $\mathbb{P}_{\tilde{D}_0|\mathcal{F}}$ to conclude the proof.[6]

As defined in (11), the $\varepsilon_{n,L}^*$ provides us with a way of bounding the total variation distance between the distribution of $\bar{D}_0$ and $\tilde{D}_0$. To see this, observe that the distributions of $\tilde{D}_0$ and $\bar{D}_0$, conditioned on $\mathcal{F}$, can be described as follows. To construct $\tilde{D}_0$, we can imagine flipping $L$ fair coins to decide the assignment of $\hat{y}_i$ in each of the $(\hat{y}_{i_{2\ell-2}} \hat{y}_{i_{2\ell}})$ pairs; if it comes up heads, we swap the observed pair $(\hat{y}_{i_{2\ell-2}} \hat{y}_{i_{2\ell}})$ and if it comes up tails we do not. The observed $(x_i, y_i)$ as well as $\hat{y}_i$ for $i \notin \mathcal{L}$ are set in $\tilde{D}_0$ as they are observed in $D_0$.

$\bar{D}_0$ is constructed similarly, but we instead flip a coin with bias $(1+r((x_{i_{2\ell-1}}, \hat{y}_{i_{2\ell-1}}), (x_{i_{2\ell}}, \hat{y}_{i_{2\ell}})))^{-1}$ to decide the assignment of $(\hat{y}_{i_{2\ell-2}} \hat{y}_{i_{2\ell}})$ – again, heads indicates that we swap the observed ordering, and tails indicates that we do not.

By construction, the distributions of $\bar{D}_0$ and $D_0$ are identical conditioned on $\mathcal{F}$, as $r((x_{i_{2\ell-1}}, \hat{y}_{i_{2\ell-1}}), (x_{i_{2\ell}}, \hat{y}_{i_{2\ell}}))$ denotes the true relative odds of observing each of the two possible $(x, \hat{y})$ pairings. In contrast, the distribution of $\tilde{D}_0$ is different, as it was sampled using the simplifying assumption (14) – in particular, $\tilde{D}_0$ is generated assuming $r((x_{i_{2\ell-1}}, \hat{y}_{i_{2\ell-1}}), (x_{i_{2\ell}}, \hat{y}_{i_{2\ell}})) = 1$!

The difference between the biases of these coins is bounded above by $\varepsilon_{n,L}^*$. We'll use this observation, along with the following lemma, to complete the proof.

**Lemma 3 (Bounding the total variation distance between i.i.d. coin flips)** *Let $i \in [L]$ index a sequence of i.i.d. coin flips $u_1 \ldots u_L$ each with bias $p_i$, and $v_1 \ldots v_L$ be a sequence of i.i.d. coin flips with bias $q_i$. Then we can show:*

$$\mathsf{TV}((u_1 \ldots u_L), (v_1 \ldots v_L)) \leq 1 - (1 - \max_i |p_i - q_i|)^L \tag{24}$$

We defer the proof of lemma 3 to Appendix D. This implies that the total variation distance between $\bar{D}_0$ and $\tilde{D}_0$ is bounded above by $1 - (1 - \varepsilon_{n,L}^*)^L$. This, along with (21), (22) and (23) concludes the proof of Theorem 1.

**Corollary 3.1 (Weaker type I error bound)**

$$\mathbb{P}(\tau \leq \alpha) \leq \alpha + \varepsilon_{n,L}^* L + \frac{1}{K+1} \tag{25}$$

Corollary 3.1 is a weaker bound than the one given in Theorem 1, but is easier to interpret and manipulate. We will make use of this fact in the following section; the proof is an immediate consequence of theorem 1 and provided in Appendix D for completeness.

## C   Proof of Theorem 2

To establish theorem 2, we will argue that $\varepsilon_{n,L}^*$ goes to 0 at a rate of $O(n^{-\frac{1}{d}})$. This implies that, provided $L = o(n^{\frac{1}{d}})$, the excess type I error established in theorem 1 is $o(1)$ as desired. To do this, we first show that each pair $(x_{i_{2\ell-1}}, x_{i_{2\ell}})$ chosen by **ExpertTest** will be close under the $\ell_2$ norm (lemmas 4 and 5 below). We then leverage the smoothness assumption (12) to demonstrate that this further implies that $\varepsilon_{n,L}^*$ concentrates around 0. For clarity we state auxiliary lemmas inline, and defer proofs to Appendix D.

**Finding pairs which are close under the $\ell_2$ norm.**

Let $M_L$ to be the set of matchings of size $L$ on $x_1 \ldots x_n$; i.e. each element of $M_L$ is a set of $L$ disjoint $(x, x')$ pairs. Let $m_L^*$ be the 'optimal' matching satisfying:

$$m_L^* \in \underset{z \in M_L}{\arg\min} \max_{(x,x') \in z} \|x - x'\|_2. \tag{26}$$

---

[6]This technique is inspired by the proof of type I error control given for the Conditional Permutation Test in Berrett et al. (2018); see Appendix A.2 of their work for details

That is, $m_L^*$ minimizes the maximum distance between any pair of observations in a mutually disjoint pairing of $2L$ observations. Let

$$d_L^* = \max_{(x,x')\in m_L^*} \|x - x'\|_2. \tag{27}$$

That is, the smallest achievable maximum $\ell_2$ distance over all matchings of size $L$. We'll first show that:

**Lemma 4 (Existence of an optimal matching)** *If $\mathcal{X} = [0,1]^d$ for some $d \geq 1$,*

$$d_{\frac{n}{4}}^* = O\left(n^{-\frac{1}{d}}\right) \tag{28}$$

*with probability* $1$.

That is, there exists a matching of size at least $\frac{n}{4}$ such the maximum pairwise distance in this matching scales like $O(n^{-\frac{1}{d}})$. Lemma 4 demonstrates the existence of a sizable matching in which the maximum pairwise distance indeed tends to $0$.[7] We next demonstrate that this approximates the optimal matching, at the cost of a factor of $2$ on $L$.

**Lemma 5 (Greedy approximation to the optimal matching)**

$$\max_{l\in[L]} \|x_{2l-1} - x_{2l}\|_2 \leq d_{2L}^* \tag{29}$$

That is, the maximum distance between any of the $L$ pairs of observations chosen by our algorithm will be no more than the maximum such distance in the optimal matching of size $2L$.

**Corollary 5.1** *For $L \leq \frac{n}{8}$, we have:*

$$\max_{l\in[L]} \|x_{2l-1} - x_{2l}\|_2 = O\left(n^{-\frac{1}{d}}\right) \tag{30}$$

This follows immediately by invoking lemma 4 to bound the right hand side of lemma 5. Corollary 5.1 demonstrates that as $n$ grows large, the maximum pairwise $\ell_2$ distance between $L$ greedily chosen pairs will go to zero at a rate of $O\left(n^{-\frac{1}{d}}\right)$ provided $L \leq \frac{n}{8}$. We now show that the smoothness condition (12) further implies that, under these same conditions, we recover the asymptotic validity guarantee (13).

**From approximately optimal pairings to asymptotic validity.**

With the previous lemmas in place, the proof of theorem 2 is straightforward. Plugging the smoothness condition (12) into the definition of the odds ratio (9) yields the following:

For all $(x_{2\ell-1}, y_{2\ell-1}), (x_{2\ell}, y_{2\ell})$,

$$r((x_{2\ell-1}, y_{2\ell-1}), (x_{2\ell}, y_{2\ell})) \in \left[\frac{1}{(1 + C\|x_{2\ell-1} - x_{2\ell}\|_2)^2}, (1 + C\|x_{2\ell-1} - x_{2\ell}\|_2)^2\right] \tag{31}$$

Where $C > 0$ is the same constant in the definition of the smoothness condition (12). Corollary 5.1 shows that $\|x_{2\ell-1} - x_{2\ell}\|_2 = O\left(n^{-\frac{1}{d}}\right)$, so (31) immediately implies that $\varepsilon_{n,L}^*$, defined in (11), also goes to zero at a rate of $O\left(n^{-\frac{1}{d}}\right)$. Thus, if we take $L$ to be a constant and $K \to \infty$, the type I error given in (10) can be rewritten as

$$\mathbb{P}\left(\tau_K \leq \alpha\right) \leq \alpha + (1 - (1 - \varepsilon_{n,L}^*)^L) + \frac{1}{K+1} \tag{32}$$

$$\leq \alpha + \varepsilon_{n,L}^* L + \frac{1}{K+1} \tag{33}$$

$$= \alpha + O\left(n^{-\frac{1}{d}}\right) \tag{34}$$

---

[7]In principle, we could find this optimal matching by binary searching for $d_L^*$ using the non-bipartite maximal matching algorithm of Edmonds (1965); for simplicity, our implementation uses a greedy matching strategy instead.

Where (33) follows from corollary 3.1. If we instead allow $L$ to scale like $o(n^{\frac{1}{d}})$ (still taking $K \to \infty$), (33) implies:

$$\mathbb{P}(\tau_K \leq \alpha) \leq \alpha + o(1) \tag{35}$$

which concludes the proof of theorem 2.

## D  Proofs of auxiliary lemmas

**Proof of Lemma 3**.

Recall that one definition of the total variation distance between two distributions $P$ and $Q$ is to consider the set of *couplings* on these distributions. In particular, the total variation distance can be equivalently defined as:

$$\mathsf{TV}(P, Q) = \inf_{(X,Y) \sim C(P,Q)} \mathbb{P}(X \neq Y) \tag{36}$$

Where $C(\cdot, \cdot)$ is the set of couplings on its arguments. Consider then the following straightforward coupling on $X := (u_1 \ldots u_L)$ and $Y := (v_1 \ldots v_L)$: draw $L$ random numbers independently and uniformly from the interval $[0, 1]$. Denote these by $c_1 \ldots c_L$. Let $u_i = \mathbb{1}[c_i \leq p_i]$, and $v_i = \mathbb{1}[c_i \leq q_i]$. It's clear that $X$ and $Y$ are marginally distributed according to $p_1 \ldots p_L$ and $q_1 \ldots q_L$, respectively. Furthermore, the probability that $u_i \neq v_i$ is $|p_i - q_i|$ by construction. Thus we have:

$$\mathbb{P}(X \neq Y) = 1 - \mathbb{P}(X = Y) = 1 - \Pi_{i \in [L]}(1 - |p_i - q_i|) \leq 1 - (1 - \max_i |p_i - q_i|)^L \tag{37}$$

This concludes the proof.

**Proof of Corollary 3.1.**

In the preceding proof of lemma 3, observe that we could have instead written:

$$\mathbb{P}(X \neq Y) = \bigcup_{i \in [L]} \{v_i \neq u_i\} \underbrace{\leq}_{\text{union bound}} \sum_{i \in [L]} |p_i - q_i| \leq L \max_{i \in [L]} |p_i - q_i| \tag{38}$$

Specializing this result to the definitions $\bar{D}_0$ and $\tilde{D}_0$ (and, in particular, the definition of $\varepsilon_{n,L}^*$) completes the proof.

**Proof of Lemma 4**.

Our proof will proceed via a covering argument. In particular, we cover the feature space $[0, 1]^d$ with a set of non-overlapping d-dimensional hypercubes, each of which has edge length $0 < b < 1$, and show that sufficiently many pairs $(x, x')$ must lie in the same 'small' hypercube. To that end, let $C = \{c_1 \ldots c_k\}$ be a set of hypercubes of edge length $b$ with the following properties:

$$\forall c \in C, c \subseteq [-b, 1 + b]^d \tag{39}$$

$$\forall c, c' \in C, c \cap c' = \emptyset \tag{40}$$

$$\forall x \in D_0, \exists c \in C \mid x \in c \tag{41}$$

Where $D_0$ is the observed data. It's clear that such a covering $C$ must exist, for example by arranging $c_1 \ldots c_k$ in a regularly spaced grid which cover $[0, 1]^d$ (though note that per condition (39), some of these 'small' hypercubes may extend outside $[0, 1]^d$ if $b$ does not evenly divide 1). Such a covering may be difficult to index as care must be exercised around the boundaries of each small hypercube; however, as we only require the existence of such a covering, we ignore these details. We now state the following elementary facts:

$$|C| \leq \lfloor \frac{(1 + 2b)^d}{b^d} \rfloor \tag{42}$$

$$\forall c \in C, x, x' \in c, ||x - x'||_2 \leq b\sqrt{d} \tag{43}$$

Where (42) follows because the volume of each $c \in C$ is $b^d$, and the total volume of all such hypercubes cannot exceed the volume of the containing hypercube $[-b, 1+b]^d$, which gives us an upper bound on the size of the cover $C$. Furthermore, (43) tells us that for any $(x, x')$ which lie in the same 'small' hypercube $c$, we have $\|x - x'\|_2 \leq b\sqrt{d}$.

Let $n_c := |\{x_i \mid x_i \in c\}|$ denote the number of observations contained in each small hypercube $c \in C$.

**Corollary 5.2** *For any $c \in C$, there exist at least $\lfloor \frac{n_c}{2} \rfloor$ disjoint pairs $(x, x') \in c$ such that $\|x - x'\|_2 \leq b\sqrt{d}$.*

With these preliminaries in place, we'll proceed to prove lemma 4. To do this, we'll first state one additional auxiliary lemma.

Let $N_{a,b} := \frac{a^d}{b^d} \geq \lfloor \frac{a^d}{b^d} \rfloor$, an upper bound on the number of non-overlapping 'small' hypercubes with edge length $b$ which can fit into $[0, a]^d$. We'll show for any $z > 0$, with $b := \frac{z}{\sqrt{d}}, a := 1 + 2b$, we have:

**Lemma 6 (Pairwise distance in terms of packing number)**

$$n \geq 2N_{a,b} \Rightarrow \exists \frac{n}{4} \text{ pairs satisfying } \|x - x'\|_2 \leq z \tag{44}$$

That is, the pairwise distance between the closest set of $\frac{n}{4}$ pairs (half the observed data in total) can be written in terms of the appropriately parameterized covering number. We defer the proof of this lemma to the following section. For now, we simply plug in the definition of $N_{a,b}$ and rearrange to recover:

$$n \geq 2N_{a,b} = 2\frac{\left(1 + 2\frac{z}{\sqrt{d}}\right)^d}{\left(\frac{z}{\sqrt{d}}\right)^d} \Rightarrow \frac{2^{\frac{1}{d}}\sqrt{d}}{n^{\frac{1}{d}} - 2^{1+\frac{1}{d}}} \leq z \tag{45}$$

Recall that $z$ is the maximum distance between any pairs $(x, x')$ contained in the same small hypercube with edge length $\frac{z}{\sqrt{d}}$. The preceding argument holds for all $z > 0$ which satisfy (45), so in particular, it holds for

$$z^* := \frac{2^{\frac{1}{d}}\sqrt{d}}{n^{\frac{1}{d}} - 2^{1+\frac{1}{d}}}. \tag{46}$$

$z^*$ is the maximum pairwise distance corresponding to one possible matching on $\frac{n}{4}$ $(x, x')$ pairs, so this further implies that there exists a matching $M$ of size $\frac{n}{4}$ such that:

$$\max_{(x,x')\in M} \|x - x'\|_2 \leq \frac{2^{\frac{1}{d}}\sqrt{d}}{n^{\frac{1}{d}} - 2^{1+\frac{1}{d}}} = O(n^{-\frac{1}{d}})$$

With probability 1. Thus, it follows that the maximum distance between any pair in the optimal matching $d^*_{\frac{n}{4}}$ also satisfies:

$$d^*_{\frac{n}{4}} = O\left(\frac{2^{\frac{1}{d}}\sqrt{d}}{n^{\frac{1}{d}} - 2^{1+\frac{1}{d}}}\right) = O\left(n^{-\frac{1}{d}}\right)$$

With probability 1, as desired. This establishes the existence of a matching of up to $L = \frac{n}{4}$ disjoint pairs $(x, x') \in [0, 1]^d$ such that the maximum distance between any such pair scales like $O\left(n^{-\frac{1}{d}}\right)$.

We also consider the case where instead of $\mathcal{X} := [0, 1]^d$, we instead have $\mathbb{P}\left(X \in [0, 1]^d\right) \geq 1 - \delta$ for some $\delta \in (0, 1)$. For example, this will capture the case where $X$ is a (appropriately re-centered and re-scaled) multivariate Gaussian. In this case, we provide a corresponding high probability version of lemma 4.

**Corollary 6.1** *Suppose instead of $\mathcal{X} := [0,1]^d$, we have for some $\delta \in (0,1)$:*

$$\mathbb{P}(X \in [0,1]^d) \geq 1 - \delta \tag{47}$$

*Define $m := (1-\delta)^2 n$*

*We can then show:*

$$\mathbb{P}\left(d^*_{\frac{m}{4}} \leq \frac{2^{\frac{1}{d}}\sqrt{d}}{m^{\frac{1}{d}} - 2^{1+\frac{1}{d}}}\right) \geq 1 - e^{-\frac{\delta^2(1-\delta)n}{2}} \tag{48}$$

That is, we can still achieve a constant factor approximation to the optimal matching in Lemma 4 with probability that exponentially approaches 1.

**Proof of Corollary 6.1**

Define the set of points which falls in $[0,1]^d$ as follows:

$$S_0 := \{X_i \mid X_i \in [0,1]^d\} \tag{49}$$

and

$$n_0 := |S_0| \tag{50}$$

It is clear that in this setting, the proof of lemma 4 holds if we simply replace $n$ with $n_0$, the realized number of observations which fall in $[0,1]^d$. However, $n_0$ is now a random quantity which follows a binomial distribution with mean $(1-\delta)n$ (recall that we assume $(x_i, y_i, \hat{y}_i)$ are drawn i.i.d. throughout). Thus, all that remains is to bound $n_0$ away from 0, which we can do via a simple Chernoff bound:

$$\mathbb{P}(n_0 \leq (1-\delta)^2 n) \leq e^{-\frac{\delta^2(1-\delta)n}{2}} \tag{51}$$

Thus, it follows that

$$\mathbb{P}(n_0 \geq (1-\delta)^2 n) \geq 1 - e^{-\frac{\delta^2(1-\delta)n}{2}} \tag{52}$$

Thus, we have shown $n_0 \geq m$ with the desired probability. It is clear that we only require a lower bound on $n_0$ to recover the result of Theorem 4, as additional observations which fall in $[0,1]^d$ can only improve the quality of the optimal matching $d^*_{\frac{m}{4}}$.

**Proof of Lemma 5**

We will show that the procedure in **ExpertTest** which greedily pairs the closest remaining pair of points $L$ times will always be able to choose at least one of the pairs in an optimal matching of size $2L$. Intuitively, this is because each pair $(x, x')$ chosen by **ExpertTest** can only 'rule out' at most two pairs $(x, x''), (x', x''')$ in any optimal matching of size $2L$. Thus, our greedy algorithm for choosing $L$ pairs can perform no worse than an optimal matching of size $2L$, the sense of minimizing the maximum pairwise distance.

Let $m^*_{2L}$ be an optimal matching of size $2L$ in the sense of (26). Then suppose towards contradiction that:

$$\max_{l \in [L]} ||x_{2l-1} - x_{2l}||_2 > d^*_{2L} \tag{53}$$

Where $d^*_{2L}$ is the smallest achievable maximum distance for any matching of size $2L$ as in (27).

Finally, let $l_m := \arg\min_{l \in [L]} ||x_{2l-1} - x_{2l}||_2 > d^*_{2L}$; i.e. the first pair which is chosen by **ExpertTest** that violates (53). Because pairs are chosen greedily to minimize $\ell_2$ distance, and $m^*_{2L}$ is a matching of size $2L$ where all pairs are separated by at most $d^*_{2L}$ under the $\ell_2$ norm, it must be that *none* of the pairs which make up $m^*_{2L}$ were available to **ExpertTest** at the $l_m$-th iteration. In particular, at least one element of every $(x, x')$ pair in $m^*_{2L}$ must have been selected on a previous iteration:

$$\forall (x, x') \in m^*_{2L}, x \in \{x_1 \ldots x_{2l_m-2}\} \vee x' \in \{x_1 \ldots x_{2l_m-2}\} \tag{54}$$

As $m_{2L}^*$ contains $2L$ disjoint pairs – $4L$ observations total – this implies that $2l_m - 2 \geq 2L \Rightarrow l_m - 1 \geq L \Rightarrow l_m > L$. This is a contradiction, as **ExpertTest** only chooses $L$ pairs, so $l_m$ only ranges in $[1, L]$. This completes the proof.

**Corollary 6.2** *Validity in finite samples*

*Theorem 2 implies that we can achieve a bound on the excess type one error in finite samples if we knew the constant $C$ in (12). In particular, let*

$$m^* := \max_{\ell \in [L]} ||x_{2\ell-1} - x_{2\ell}||_2 \tag{55}$$

$$\epsilon^* := \max_{r \in [(1+Cm^*)^{-2}, (1+Cm^*)^2]} \left| \frac{1}{r+1} - \frac{1}{2} \right| \tag{56}$$

*Then (10) implies that we can always construct a valid (if underpowered) test at exactly the nominal size $\alpha$ by updating our `REJECT` threshold to*

$$\alpha - \left(1 - (1 - \epsilon^*)^L\right) - \frac{1}{K+1}$$

**Proof of lemma 6**

let $C := \{c_1 ... c_k\}$ denote any set of $k$ 'small' nonoverlapping hypercubes of edge length $b$ satisfying properties (39), (40) and (41). As discussed in the proof of lemma 4, each element of $C$ is not guaranteed to lie strictly in $[0, 1]$. Rather, each $c \in C$ must merely intersect $[0, 1]^d$, implying that each element of the cover is instead contained in the slightly larger hypercube $[-b, 1 + b]^d$. As in the proof of lemma 4, we'll again let $n_c$ denote the number of observations $x_i$ which lie in some $c \in C$.

By Corollary 5.2, we have that $\lfloor \frac{n_c}{2} \rfloor$ pairs in each $c \in C$ will satisfy $||x - x'||_2 \leq b\sqrt{d} = z$. Thus what's left to show is that:

$$n \geq 2N_{a,b} \Rightarrow \sum_{j \in [k]} \lfloor \frac{n_{c_j}}{2} \rfloor \geq \frac{n}{4}$$

We can see this via the following argument:

$$\sum_{j \in [k]} \lfloor \frac{n_{c_j}}{2} \rfloor \geq \sum_{j \in [k]} \left( \frac{n_{c_j}}{2} - \frac{1}{2} \right) \tag{57}$$

$$= \frac{n}{2} - \frac{k}{2} \tag{58}$$

$$\geq \frac{n}{2} - \frac{N_{a,b}}{2} \tag{59}$$

$$\geq \frac{n}{2} - \frac{n}{4} = \frac{n}{4} \tag{60}$$

Where (59) follows from (42) and the definition of $N_{a,b}$, and (60) follows because $n \geq 2N_{a,b}$ by assumption. This completes the proof.

# E Omitted Details from Section 5

## E.1 Identifying relevant patient encounters and classifying outcomes

As described in Section 5, we consider a set of 3617 patients who presented with signs or symptoms of acute gastrointestinal bleeding at the emergency department at a large quaternary academic hospital system from January 2014 to December 2018. These patient encounters were identified using a database mapping with a standardized ontology (SNOMED-CT) and verified by manual physician chart review. Criteria for inclusion were the following: any text that identifies acute gastrointestinal bleeding for hematemesis, melena, hematochezia from either patient report or physical exam findings

(which were considered equally valid for the purposes of inclusion). Exclusion criteria were the following: patients with other reasons for overt bleeding symptoms (e.g. epistaxis) or missingness in input variables required to calculate the Glasgow-Blatchford Score. As mentioned in Section 5, only upper gastrointestinal bleeding guidelines recommend the Glasgow-Blatchford Score for routine clinical use. However, the Glasgow-Blatchford Score has also been validated for use in patients with lower gastrointestinal bleeding, and in our setting the GBS was applied to patients who presented with signs of symptoms of either upper or lower gastrointestinal bleeding. We refer the reader to Section 5 for additional details and references.

This identified a set of 3627 patients, of which a further 10 were removed from consideration due to unclear emergency department disposition (neither `Admit` nor `Discharge`). As described in Section 5, we record an adverse outcome ($Y = 1$) for admitted patients who required some form of hemostatic intervention (excluding a diagnostic endoscopy or colonoscopy), or patients who are readmitted or die within 30 days. We record an outcome of $0$ for all other patients.

The use of readmission as part of the adverse event definition is subject to two important caveats. First, we are only able to observe patients who are readmitted within the *same* hospital system. Thus, although the hospital system we consider is the dominant regional health care network, it is possible that some patients subseqeuently presented elsewhere with signs or symptoms of AGIB; such patients would be incorrectly classified as not having suffered an adverse outcome. Second, we only record an outcome of 1 for patients who are readmitted with signs or symptoms of AGIB, subject to the same inclusion criteria defined above. Patients who are readmitted for other reasons are not recorded as having suffered an adverse outcome.

### E.2 The special case of binary outcomes and predictions

In our experiments we define the loss measure $F(D) := \frac{1}{n} \sum_i \mathbb{1}[y_i \neq \hat{y}_i]$, but it's worth remarking that this is merely one choice within a large class of natural loss functions for which **ExpertTest** produces identical results when $Y, \hat{Y}$ are binary. In particular, observe that a swap of $(y_1, \hat{y}_1), (y_2, \hat{y}_2)$ can only change the value of $F(\cdot)$ if $y_1 \neq y_2$ and $\hat{y}_1 \neq \hat{y}_2$ (we'll assume throughout that all observations contribute equally to the loss; i.e. it is invariant to permutations of the indices $i \in [n]$). This implies that there are only $2^2$ out of $2^4$ possible configurations of $(y_1, \hat{y}_1, y_2, \hat{y}_2)$ where a swap can change the loss at all. Of these, two configurations create a false negative and a false positive in the synthetic data which did not exist in the observed data:

$$\underbrace{(y_1 = 1, \hat{y}_1 = 1, y_2 = 0, \hat{y}_2 = 0)}_{\text{original data}} \underset{\text{swap}}{\rightarrow} \underbrace{(y_1 = 1, \hat{y}_1 = 0, y_2 = 0, \hat{y}_2 = 1)}_{\text{synthetic data}}$$

$$(y_1 = 0, \hat{y}_1 = 0, y_2 = 1, \hat{y}_2 = 1) \underset{\text{swap}}{\rightarrow} (y_1 = 0, \hat{y}_1 = 1, y_2 = 1, \hat{y}_2 = 0)$$

The other two configurations which change the loss are symmetric, in that a swap *removes* both a false negative and false positive that exists in the observed data:

$$(y_1 = 0, \hat{y}_1 = 1, y_2 = 1, \hat{y}_2 = 0) \underset{\text{swap}}{\rightarrow} (y_1 = 0, \hat{y}_1 = 0, y_2 = 1, \hat{y}_2 = 1)$$

$$(y_1 = 1, \hat{y}_1 = 0, y_2 = 0, \hat{y}_2 = 1) \underset{\text{swap}}{\rightarrow} (y_1 = 1, \hat{y}_1 = 1, y_2 = 0, \hat{y}_2 = 0)$$

Thus, for any natural loss function which is strictly increasing in the number of mistakes $\sum_i \mathbb{1}[y_i \neq \hat{y}_i]$, the first two configurations of $(y_1, \hat{y}_1, y_2, \hat{y}_2)$ will induce swaps which strictly increase the loss, while the latter two will induce swaps that strictly decrease the loss. This means that for a given set of $L$ pairs, we can compute the number of swaps which would increase (respectively, decrease) the loss for *any* function in this class of natural losses. In particular, this class includes loss functions which may assign arbitrarily different costs to false negatives and false positives. Thus, in the particular context of assessing physician triage decisions, our results are robust to variation in the way different physicians, patients or other stakeholders might weigh the relative cost of false negatives (failing to hospitalize patients who should have been admitted) and false positives (hospitalizing patients who could have been discharged to outpatient care).

## E.3  Alternative feature spaces

In Section 5, we run **ExpertTest** by pairing patients who appear as close as possible (and, in most cases, identical) with respect to the nine features which are provided as input to the Glasgow-Blatchford score. While this test has perhaps the most natural interpretation in our setting, we could in general choose other ways of representing patients. In particular, a natural question to ask is whether physicians successfully distinguish patients who present with identical Glasgow-Blatchford scores (or, more generally, with identical predictions under some model of interest). That is, rather than letting $X \in [0, 1]^9$ be the nine patient characteristics which are provided as *input* to the GBS, we instead let $X \in \{0, 1 \ldots 23\}$ be the GBS itself. While this is in some sense 'weaker' than the test we run in Section 5 – the GBS is a deterministic function of the richer feature space we chose there – recall that the validity of **ExpertTest** relies on finding pairs which are identical or nearly identical in the feature space. Thus, choosing to instead condition on a simple univariate prediction can in general yield useful insight when **ExpertTest** fails to offer conclusive results in the 'natural' input space. We present the results of this experiment in Table 3 below.

| L | mismatched pairs | swaps that increase loss | swaps that decrease loss | $\tau$ |
|---|---|---|---|---|
| 100 | 0 | 4 | 0 | 0.043 |
| 250 | 0 | 11 | 2 | 0.004 |
| 500 | 0 | 21 | 3 | <.001 |
| 1000 | 0 | 36 | 3 | <.001 |
| 1808 | 5 | 71 | 5 | <.001 |

Table 3: The results of running ExpertTest, where each pair of patients is chosen to be as similar as possible with respect to their Glasgow-Blatchford scores. $L$ indicates the number of pairs selected for the test, of which 'mismatched pairs' are not identical to each other. Swaps that decrease (respectively, increase) loss indicates how many of the $L$ pairs result in a decrease (respectively, increase) in the 0/1 loss when their corresponding hospitalization decisions are exchanged with each other. $\tau$ is the p-value obtained from running ExpertTest.

As we can see, we get very similar results to those presented in Section 5. Consistent with Table 2, we find no mismatched pairs for $L \leq 1000$, and we again obtain the smallest possible p-values of $\frac{1}{K+1} = \frac{1}{1001}$ for larger values of $L$.

Another experiment we might run is to condition on a *richer* feature space than the one we chose in Section 5. In particular, recall that of the nine inputs to the GBS, four of them (blood urea nitrogen (BUN), hemoglobin (HGB), systolic blood pressure (SBP) and pulse) are real-valued test scores or vital signs. Part of the standard construction of the Glasgow-Blatchford Score entails converting these features to simple discrete scales, with larger values indicating that a patient is higher risk. A useful consequence of this convention is that it allows us to find large numbers of patients who appear identical with respect to these discretized features. However, we can also run **ExpertTest** in the 'original' feature space, where each of these four characteristics take values in $\mathbb{R}$ and thus effectively guarantee that we fail to find pairs of patients who are exactly identical. We present the results of this test (where the only transformation we apply is to again rescale each feature to lie in $[0, 1]$) in Table 4 below.

Here we again see strong evidence of expertise, with p-values equal to $\frac{1}{K+1} = \frac{1}{1001}$ for larger values of $L$. However, unlike in Table 2 and Table 3, we now fail to find identical pairs of patients for *any* value of $L$. This raises the possibility that **ExpertTest** is not valid in this setting, as type I error control now relies on the value of $\varepsilon_{n,L}^*(11)$. While we cannot compute $\varepsilon_{n,L}^*$ without making assumptions about the underlying data distribution, we can provide a sense check by examining the Euclidian distance between each pair of patients. If it is very close to $0$ for most or all pairs, we might reasonably expect that **ExpertTest** recovers approximate type I error control. We present these results in Figure 1 below.

As expected, we fail to find any pairs of patients who are identical in this partially real-valued feature space. Nonetheless, we see that for $L \leq 1000$, every pair is chosen such that the Euclidian distance between them is very close to $0$, which suggests that **ExpertTest** is likely to be approximately valid for these experiments. We also see however that for $L = 1808$, some pairs are chosen such that the

| L | mismatched pairs | swaps that increase loss | swaps that decrease loss | $\tau$ |
|---|---|---|---|---|
| 100 | 100 | 5 | 0 | 0.015 |
| 250 | 250 | 13 | 2 | 0.005 |
| 500 | 500 | 23 | 3 | <.001 |
| 1000 | 1000 | 36 | 6 | <.001 |
| 1808 | 1808 | 66 | 6 | <.001 |

Table 4: The results of running ExpertTest, where each pair of patients is chosen to be as similar as possible with respect to nine patient characteristics. The Glasgow-Blatchford score is computed from these nine characteristics, after a pre-processing step which discretizes each feature. $L$ indicates the number of pairs selected for the test, of which 'mismatched pairs' are not identical to each other. Swaps that decrease (respectively, increase) loss indicates how many of the $L$ pairs result in a decrease (respectively, increase) in the 0/1 loss when their corresponding hospitalization decisions are exchanged with each other. $\tau$ is the p-value obtained from running ExpertTest.

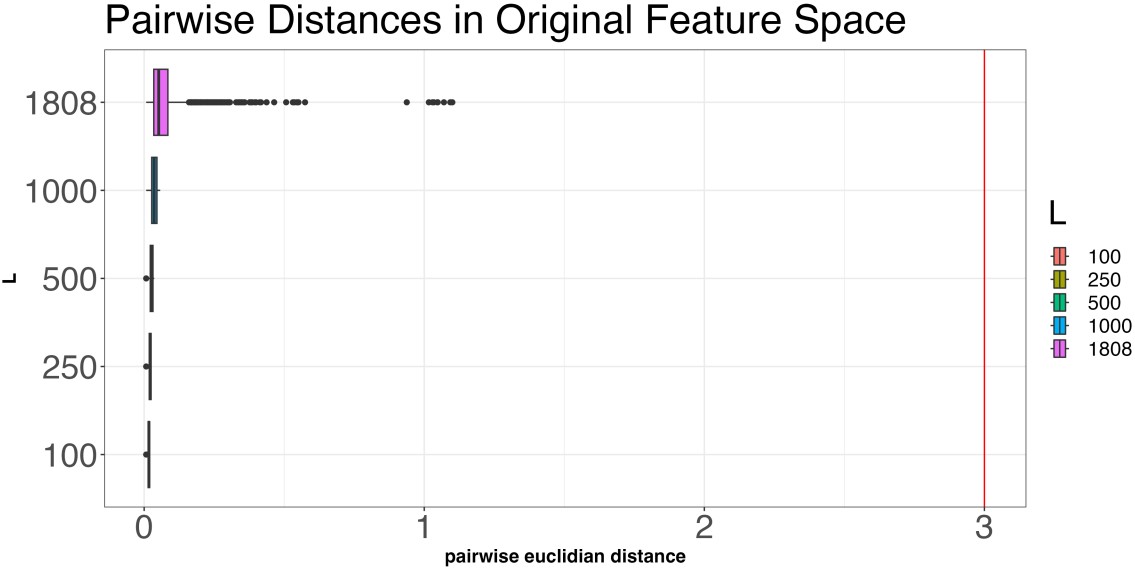

Figure 1: Distribution of Euclidian distances between each pair of patients chosen by **ExpertTest** when patients are represented as a vector of nine patient characteristics, of which four – blood urea nitrogen (BUN), hemoglobin (HGB), systolic blood pressure (SBP) and pulse – are real-valued. $L$ indicates the number of pairs of patients chosen for each experiment, with the boxplot indicating the distribution of pairwise Euclidian distances between them. The red line at $\sqrt{9} = 3$ indicates the maximum possible Euclidian distance in this feature space.

pairwise distance is more than $30\%$ of its maximum possible value, suggesting that **ExpertTest** does indeed incur substantial excess type I error when $L = 1808$.

## F    Numerical Experiments

We first elaborate here on the example 1 presented in the introduction. Consider the following stylized data generating process:

**Example: experts can add value despite poor performance.**

Let $X, U, \epsilon_1, \epsilon_2$ be independent random variables distributed as follows:

$$X \sim \mathcal{U}([-2, 2]), U \sim \mathcal{U}([-1, 1]), \epsilon_1 \sim \mathcal{N}(0, 1), \epsilon_2 \sim \mathcal{N}(0, 1)$$

Where $\mathcal{U}(\cdot)$ and $\mathcal{N}(\cdot, \cdot)$ are the uniform and normal distribution, respectively. Suppose the true data generating process for the outcome of interest $Y$ is

$$Y = X + U + \epsilon_1$$

Suppose a human expert constructs a prediction $\hat{Y}$ which is intended to forecast $Y$ and can be modeled as:

$$\hat{Y} = \text{sign}(X) + \text{sign}(U) + \epsilon_2$$

Where $\text{sign}(X) := \mathbb{1}[X > 0] - \mathbb{1}[X < 0]$.

We compare this human prediction to that of an algorithm $\hat{f}(\cdot)$ which can only observe $X$, and correctly estimates

$$\hat{f}(X) = \mathbb{E}[Y \mid X] = X$$

As described in the introduction, we use this example to demonstrate that **ExpertTest** can detect that the forecast $\hat{Y}$ is incorporating the unobserved $U$ even though the accuracy of $\hat{Y}$ is substantially worse than that of $\hat{f}(X)$. In particular, we consider the *mean squared error* (MSE) of each of these predictors:

$$\text{Algorithm MSE} := \frac{1}{n} \sum_i (Y_i - \hat{f}(X_i))^2$$

$$\text{Human MSE} := \frac{1}{n} \sum_i (Y_i - \hat{Y}_i)^2$$

We'll show below that the Algorithm MSE is substantially smaller than the Human MSE. However, we may also wonder whether the performance of the human forecast $\hat{Y}$ is somehow artificially constrained by the the relative scale of $\hat{Y}$ and $Y$, as the $\text{sign}(\cdot)$ operation restricts the range of $\hat{Y}$. For example, a forecaster who always outputs $\hat{Y} = \frac{Y}{100}$ is perfectly correlated with the outcome but will incur very large squared error; this is a special case of the more general setting where human forecasts are directionally correct but poorly *calibrated*. To test this hypothesis, we can run ordinary least squares regression (OLS) of $Y$ on $\hat{Y}$ and compute the squared error of this rescaled prediction. It is well known OLS estimates the optimal linear rescaling with respect to squared error, and we further use the *in sample* MSE of this rescaled prediction to provide a lower bound on the achievable loss. In particular, let:

$$(\beta^*, c^*) := \min_{\beta, c \in \mathbb{R}} ||Y - \beta\hat{Y} - c||_2^2 \tag{61}$$

$$\text{Rescaled Human MSE} := \frac{1}{n} \sum_i (Y_i - \beta^*\hat{Y}_i - c^*)^2 \tag{62}$$

In Table 5 we report the mean squared error (plus/minus two standard deviations) over 100 draws of $n = 1000$ samples from the data generating process described above. As we can see, both the original and rescaled human forecasts substantially underperform $\hat{f}(\cdot)$.

Table 5: Expert vs Algorithm Performance

| Algorithm MSE | Human MSE | Rescaled Human MSE |
|---|---|---|
| 1.33 ± 0.12 | 2.67 ± 0.24 | 1.92 ± 0.16 |

We now assess the power of **ExpertTest** in this setting by repeatedly simulating $n = 1000$ draws of $(X, U, \epsilon_1, \epsilon_2)$ along with the associated outcomes $Y := X + U + \epsilon_1$ and expert predictions $\hat{Y} := \text{sign}(X) + \text{sign}(U) + \epsilon_2$. We sample 100 datasets in this manner, and run **ExpertTest** on each one with $L, K = 100$, and the distance metric $m(x, x') := \sqrt{(x - x')^2}$. The distribution of p-values $\tau_1...\tau_{100}$ is plotted in Figure 2.

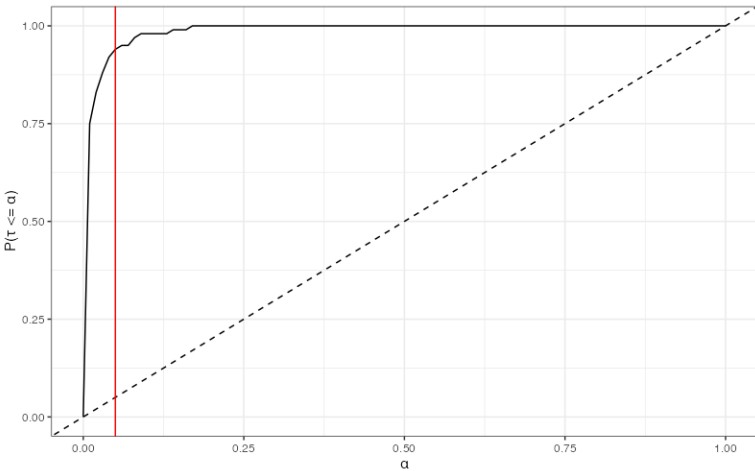

Figure 2: distribution of $\tau$ is sharply nonuniform when the expert incorporates unobserved information $U$ in the toy example. The vertical red line indicates a critical threshold of $\alpha = .05$, and the dashed line traces a uniform distribution.

We see that **ExpertTest** produces a highly nonuniform distribution of the p-value $\tau$, and rejects the null hypothesis $94\%$ of the time at a critical value of $\alpha = .05$. To assess whether this power comes at the expense of an inflated type I error, we also run **ExpertTest** with both $X$ and $U$ 'observed'; in particular, suppose the distance measure was instead $m((x, u), (x', u')) = \sqrt{(x - x')^2 + (u - u')^2}$ with everything else defined as above. The distribution of $\tau$ in this setting is again plotted in Figure 3.

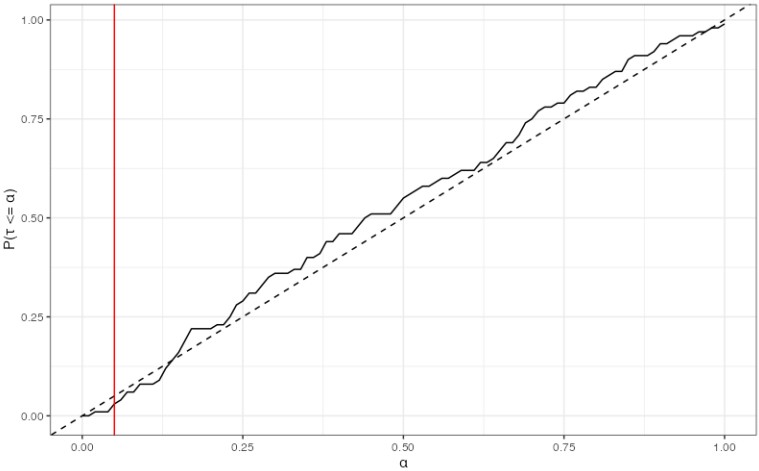

Figure 3: distribution of $\tau$ is approximately uniform when the expert does not incorporate unobserved information in the toy example. The vertical red line indicates a critical threshold of $\alpha = .05$, and the dashed line traces a uniform distribution.

When both $X$ and $U$ are observed, and thus the null hypothesis should not be rejected, we instead see that we instead get an approximately uniform distribution of $\tau$ with a false discovery rate of only .03 at a critical value of $\alpha = .05$. Thus, the power of **ExpertTest** to detect that the synthetic expert is incorporating some unobserved information $U$ does not come at the expense of inflated type I error, at least in this synthetic example.

**Assessing the power of ExpertTest**

We now present additional simulations to highlight how the power of **ExpertTest** scales with the number of pairs $L$ and the sample size $n$ in a more general setting. In particular, we consider a simple

synthetic dataset $(x_i, y_i, \hat{y}_i), i \in [n] \equiv \{1, \ldots, n\}$ where $x_1 \ldots x_n = [1, 1, 2, 2, \ldots \frac{n}{2}, \frac{n}{2}]'$ and $y_1 \ldots y_n$ is the alternating binary string $[0, 1, 0, 1 \ldots 0, 1]'$ (we consider only even $n$ for simplicity). This guarantees that each of the $L$ pairs chosen are such that $(x_{2\ell-1} = x_{2\ell})$ and $y_{2\ell-1} \neq y_{2\ell}$. Importantly, it's also clear that $x$ is uninformative about the true outcome $y$ – if the expert can perform better than random guessing, it must be by incorporating some unobserved signal $U$.

We model this unobserved signal by an 'expertise parameter' $\delta \in [0, \frac{1}{2}]$. In particular, for each pair $(y_{2\ell-1}, y_{2\ell})$ for $\ell \in [1 \ldots \frac{n}{2}]$, we sample $(\hat{y}_{2\ell-1}, \hat{y}_{2\ell})$ such that $(\hat{y}_{2\ell-1}, \hat{y}_{2\ell}) = (y_{2\ell-1}, y_{2\ell})$ with probability $\frac{1}{2} + \delta$ and $(y_{2\ell}, y_{2\ell-1})$ otherwise. Intuitively, $\delta$ governs the degree to which the expert predictions $\hat{Y}$ incorporate unobserved information – at $\delta = 0$, we model an expert who is randomly guessing, whereas at $\delta = \frac{1}{2}$ the expert predicts the outcome with perfect accuracy.

First, we consider the case of $n \in \{200, 600, 1200\}$ and fix $L$ at $\frac{n}{8}$ as suggested by the proof of Theorem 2. For each of these cases, we examine how the discovery rate scales with the expertise parameter $\delta \in \{0, .05 \ldots .45, .50\}$. In particular, we choose a critical threshold of $\alpha = .05$ and compute how frequently **ExpertTest** rejects $H_0$ over 100 independent draws of the data for each value of $\delta$. These results are plotted below in Figure 4.

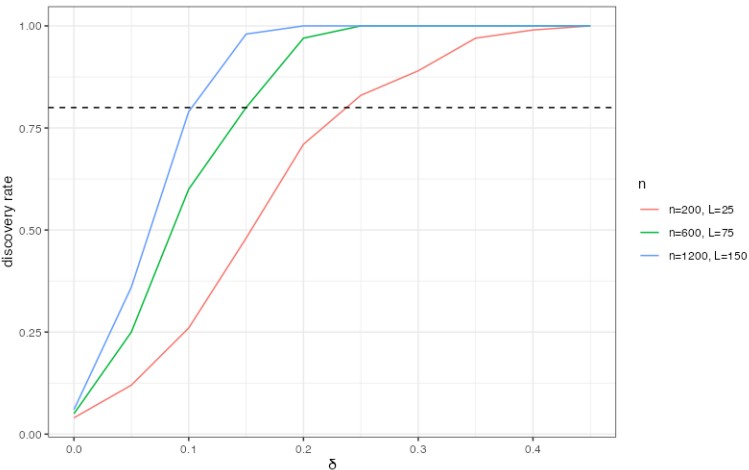

Figure 4: The power of **ExpertTest** as a function of sample size $n$ and expertise parameter $\delta$. The horizontal dashed line corresponds to a power of $80\%$

Unsurprisingly, the power of **ExpertTest** depends critically on the sample size – at $n = 1200$, **ExpertTest** achieves $80\%$ power in rejecting $H_0$ when the expert only performs modestly better than random guessing ($\delta \approx .1$). In contrast, at $n = 200$, **ExpertTest** fails to achieve $80\%$ power until $\delta \approx .25$ – corresponding to an expert who provides the correct predictions over $75\%$ of the time even when the observed $x$ is completely uninformative about the true outcome.

Next we examine how the power of **ExpertTest** scales with $L$. We now fix $n = 600$ and let $\delta = .2$ to model an expert who performs substantially better than random guessing, but is still far from providing perfect accuracy. We then vary $L \in \{20, 40 \ldots 200\}$ and plot the discovery rate (again at a critical value of $\alpha = .05$, over 500 independent draws of the data) for each choice of $L$. These results are presented below in Figure 5.

As expected, we see that power is monotonically increasing in $L$, and asymptotically approaching 1. With $\delta = .2$, we see that **ExpertTest** achieves power in the neighborhood of only $50\%$ with $L = 20$ pairs, but sharply improves to approximately $80\%$ power once $L$ increases to 40. Beyond this threshold we see that there are quickly diminishing returns to increasing $L$.

**Excess type I error of ExpertTest**

Recall that, per Theorem 1, **ExpertTest** becomes more likely to incorrectly reject $H_0$ as $L$ increases relative to $n$. In particular, larger values of $L$ will force **ExpertTest** to choose $(x, x')$ pairs which are farther apart under any distance metric $m(\cdot, \cdot)$, and thus induce larger values of $\varepsilon_{n,L}^*$ as defined in

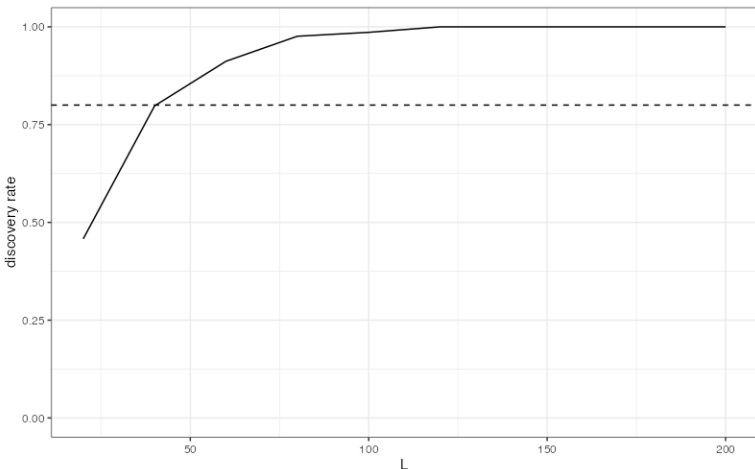

Figure 5: The power of **ExpertTest** as a function of $L$, with $n = 600, \delta = .2$. The horizontal dashed line corresponds to a power of $80\%$

(11). Furthermore, even for fixed $\varepsilon^*_{n,L} > 0$, the type one error bound given in Theorem 1 degrades with $L$. We empirically investigate this phenomenon via the following numerical simulation.

First, let $X = (X_1, X_2, X_3) \subset \mathbb{R}^3$ be uniformly distributed over $[0, 10]^3$. Let $Y = X_1 + X_2 + X_3 + \epsilon_1$ and $\hat{Y} = X_1 + X_2 + X_3 + \epsilon_2$, where $\epsilon_1, \epsilon_2$ are independent standard normal random variables. In this setting, it's clear that $H_0 : Y \perp\!\!\!\perp \hat{Y} \mid X$ holds.

We repeatedly sample $n = 500$ independent observations from this distribution over $(X, Y, \hat{Y})$ and run **ExpertTest** for each $L \in \{25, 50 \ldots 250\}$. We let $K = 50$ and $m(x, x') := ||x - x'||_2^2$ be the $\ell_2$ distance. We let the loss function $F(\cdot)$ be the mean squared error of $\hat{Y}$ with respect to $Y$. For each scenario we again choose a critical threshold of $\alpha = .05$, and report how frequently **ExpertTest** incorrectly rejects the null hypothesis over 50 independent simulations in Figure 6.

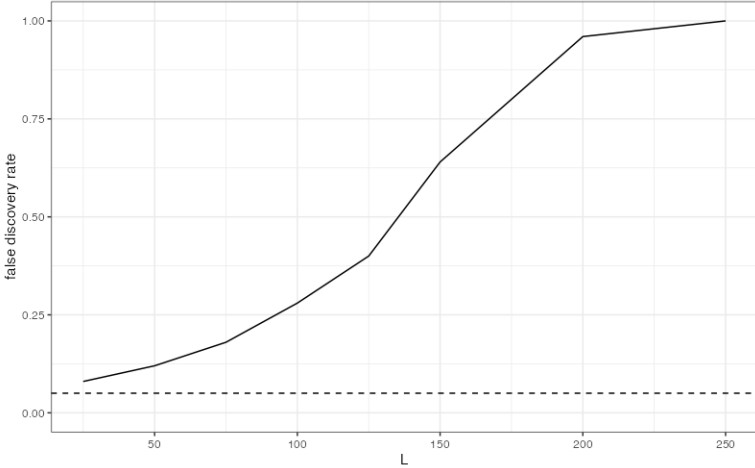

Figure 6: The type I error rate of **ExpertTest** as a function of $L$, with $n = 500$ and a critical threshold of .05. The horizontal dashed line corresponds to the nominal false discovery rate of .05

As we can see, the type I error increases sharply as a function of $L$, and **ExpertTest** incurs a false discovery rate of $100\%$ at the largest possible value of $L = \frac{n}{2}$! This suggests that significant care should be exercised when choosing the value of $L$, particularly in small samples, and responsible use of **ExpertTest** will involve leveraging domain expertise to assess whether the pairs chosen are indeed 'similar' enough to provide type I error control.

# G    Pseudocode for ExpertTest

In this section we provide pseudocode for **ExpertTest**. Inputs $D_0, L, K, \alpha, F(\cdot), m(\cdot, \cdot)$ are as defined in Section 3.

---

**ExpertTest**$(D_0, L, K, \alpha, F(\cdot), m(\cdot, \cdot))$

---

$X_0 \leftarrow \{x \mid (x, \cdot, \cdot) \in D_0\}$         ▷ initialize set of remaining observations
$P \leftarrow \emptyset$         ▷ initialize set of paired predictions

**for** $\ell = 1 : L$ **do**
     $(x_{2\ell-1}, x_{2\ell}) \leftarrow \underset{(x, x')}{\operatorname{argmin}} \, m(x, x')$      ▷ find closest remaining pair, breaking ties arbitrarily
     $X_\ell \leftarrow X_{\ell-1} \setminus \{x_{2\ell-1}, x_{2\ell}\}$
     $P \leftarrow P \cup \{(\hat{y}_{2\ell-1}, \hat{y}_{2\ell})\}$      ▷ save predictions associated with closest remaining pair
**end for**

$f_0 \leftarrow F(D_0)$         ▷ calculate observed loss

**for** $k = 1 : K$ **do**
     $D_k \leftarrow \operatorname{swap}(D_0, P, \frac{1}{2})$      ▷ independently swap each $(\hat{y}_{2\ell-1}, \hat{y}_{2\ell}) \in P$ with equal probability
     $f_k \leftarrow F(D_k)$      ▷ calculate synthetic loss
**end for**

$\tau \leftarrow \frac{1}{K} \sum_k \mathbb{1}[f_k \lesssim f_0]$      ▷ calculate quantile of observed loss, breaking ties at random

**if** $\tau \leq \alpha$ **then**      ▷ if $\tau \leq \alpha$, $H_0$ is rejected with p-value $\alpha + \frac{1}{K+1}$
     REJECT
**end if**

---

