# OpenReview forum: "Auditing for Human Expertise"
_NeurIPS.cc/2023/Conference — NeurIPS 2023 spotlight_

### Official Review · Reviewer_NoBC · 2023-06-27

**Soundness:** 4 excellent
**Presentation:** 4 excellent
**Contribution:** 3 good
**Rating:** 7
**Confidence:** 3

**Summary:**

While humans and machines oftentimes make differing decisions, it's unclear whether humans make these decisions based on extra factors or information unavailable to machines. To understand this situation, the paper proposes a statistical test to determine if expert predictions are independent of the labels, when accounting for the input features. This idea indicates whether humans rely on different information, and in a sense add additional value unseen by a model. To evaluate this, the paper analyzes doctor predictions in a hospital admitance system, and find that doctors tend to use additional information.

**Strengths:**

1. Statistical approach is well motivated and clearly demonstrates how to test for humans relying on extra information
2. Test allows for flexibility due to choice of L, allowing for different statistical properties
3. Method is evaluated on a real-world hospital dataset, and the connection between the evaluation and the methodology is clear


**Weaknesses:**

1. Method relies on dataset containing pairs that are close in input space, yet distinct in feature space; such a situation might not be generalizable
2. Evaluation is only done on one real-world dataset; a controlled evaluation of the test would give better insights into how the test performs and the impact of different parameters


**Questions:**

1. What are other choices for the function F, and would they be equally valid?
2. For the experiment with hospital admission data, would the results change if instead of using the GBS score, the individual factors that were used as inputs to the score were used?
3. In table 2, what does mismatched pairs mean?


**Limitations:**

Authors discuss limitations fairly thoroughly.

---

> ### Author Rebuttal · Authors · 2023-08-08
>
> Thank you for your feedback! Our responses are provided below.
>
> **In response to**
> > "What are other choices for the function F, and would they be equally valid?"
>
> As noted in Section 3, our test is valid for any choice of $F(\cdot)$. However, the *power* of the test will certainly depend on the choice of $F(\cdot)$. As is typical for hypothesis tests, the degree of this dependence is hard to characterize analytically since it depends on the specific distribution of $(X, Y, \hat{Y})$. Our particular choice is a natural one however, as it is well-powered to detect the specific form of dependence we care about -- whether or not the unobservables $U$ actually improve human accuracy with respect to a known measure of accuracy. Please also see our response to 7cTX for additional discussion regarding the *choice* of loss function.
>
> **In response to**
> > "In table 2, what does mismatched pairs mean?"
>
> "Mismatched pairs" refers to the number of pairs of observations chosen by ExpertTest which are not exactly identical. In the context of our case study, it means that we chose a pair of patients who did not have identical Glasgow-Blatchford scores, which risks incurring additional type I error as described in Section 4. The definition of "mismatched pairs" is given in Section 5, but we will clarify this in our final draft by including a description of each column in the table caption.
>
> **In response to**
> > "For the experiment with hospital admission data, would the results change if instead of using the GBS score, the individual factors that were used as inputs to the score were used?"
>
> Excellent question -- please see our response to 7cTX for a detailed discussion of this point. The results of this experiment are included in the attached figures.
>
> **In response to**
> > "Evaluation is only done on one real-world dataset; a controlled evaluation of the test would give better insights into how the test performs and the impact of different parameters"
>
> This concern is addressed in our response to all reviewers (see above).

---

> > ### Comment · Reviewer_NoBC · 2023-08-10
> > **Clarification**
> >
> > Thank you for your response. I have re-read the Appendix, and the synthetic experiments there are indeed what I was looking for, so thank you. I wanted a little bit more clarification on weakness #1 (whether data points that are close in feature space but far in label space exist). You answered this a bit in the general rebuttal, but I was wondering if you clarify why this generalizes?

---

> > > ### Author Response · Authors · 2023-08-10
> > >
> > > Thank you for your response. First, we should note that our test recovers (effectively) exact type I error control whenever we succeed in finding pairs of identical observations (by Theorem 1 and the definition of $\varepsilon_{n,L}^*$); this is true in both low and high dimensional feature spaces and does not rely on assumptions about the smoothness of the distribution. Of course, such pairs are less likely to exist if we are only given finite samples in high dimensional feature spaces (and occur with probability 0 in continuous feature spaces), at which point we assume that nearby observations in the feature space induce similar distributions over the forecast $\hat{Y}$ (“generative model (12)” in Section 4). This assumption is motivated by the natural intuition that humans are unlikely to finely distinguish between very similar observations, particularly in high dimensional spaces. We do not however have a way of verifying this assumption, and you are correct that we cannot make any guarantees about type I error if (1) we fail to find identical pairs and (2) the distribution does not satisfy this smoothness assumption. The fact that we need such an assumption is not surprising, as conditional independence testing is intractable in full generality (see [1] below).
> > >
> > > That said, there is no reason to believe that our assumption is *less* likely to hold in high dimensions; indeed, to take 7Ctt’s example, images which are very similar at a pixel level may look *identical* to a human expert. Furthermore, images which are quite different in pixel space may *also* induce similar distributions over forecasts, if these images are similar in some lower dimensional latent space which is relevant for human decision making. However, we may not necessarily know what the correct latent space is, or whether one exists, and thus we can only guarantee type I error control (assuming we fail to find identical pairs of observations) if a given forecasting task satisfies our modeling assumptions.
> > >
> > > [1] Shah and Peters, 2018. “The Hardness of Conditional Independence Testing and the Generalised Covariance Measure”

---

> > > > ### Comment · Reviewer_NoBC · 2023-08-11
> > > > **Response**
> > > >
> > > > Thank you for your response. I have raised the score from 6 to 7 due to these clarifications.

---

### Official Review · Reviewer_7Ctt · 2023-07-04

**Soundness:** 4 excellent
**Presentation:** 4 excellent
**Contribution:** 3 good
**Rating:** 7
**Confidence:** 3

**Summary:**

This paper proposes a statistical framework for measuring whether, in the context of algorithmic predictions, human experts incorporate valuable information in their decision making that is unknowable to the algorithm. The authors formalize this question as a simple hypothesis test: “are human expert predictions independent from the outcome variable, when conditioned on the feature vector”.  The proposed statistical test for “presence of human expertise” is straightforward, drawing on prior literature on testing conditional independence, and provides interpretable p-values. The paper then uses this framework to analyze real-world medical data from an emergency department of a large hospital, showing that physicians do in fact incorporate information above and beyond that captured by a standard algorithmic screening tool.

**Strengths:**

The main strengths of this paper are in it’s simplicity, clear exposition, and scoping of a well-defined problem that it tries to solve. The authors’ give an elegant framework for formally defining the problem of measuring valuable human expertise and their proposed test is quite intuitive. The paper presents all its ideas in a concise manner and also discusses its limitations quite candidly.

I also like the thoroughness of the experiment conducted by the authors—it seems well executed and the results are compelling evidence for the validity of their statistical test.

**Weaknesses:**

Some weaknesses of this work:

- From an algorithmic/technical standpoint, this paper uses a straightforward notion of conditional independence to define the problem, and a simple binned conditional independence test to solve it. This simplicity is not a bad thing, but it is worth noting that the main contribution of this work doesn’t present a novel technique or technical insight.
- The results and experiments of this paper would likely not extend to a high-dimensional setting. Their current experiment uses discrete, scalar features. The authors discuss this in their limitations. A concrete example: how would a test like this work for, say, radiology images, where the human predictive distribution is likely not smooth w.r.t the $\ell_2$ metric.
- I understand it’s not an easy task, but this paper would be much stronger if there were additional experiments where this technique was employed to understand the interplay between human and algorithmic decision making.

**Questions:**

- In Section 2, first paragraph: why can you assume X is independent of U, without a loss of generality?
- I would be interested in seeing this used in settings where algorithmic predictions are close to replacing human predictions. Radiology images are the immediate example that comes to mind.
- Could your test (or a small modification) detect if human experts are using additional information in a negative way? For example, could you use it to detect biasedness in college admissions or medical treatment choices because the expert is relying on information outside of the set of features they should be looking at?

**Limitations:**

Yes, good discussion of limitations and prior work.

---

> ### Author Rebuttal · Authors · 2023-08-08
>
> Thank you for your feedback! Our responses are provided below.
>
> **In response to**
> >  "could your test (or a small modification) detect if human experts are using additional information in a negative way?''
>
> 1. Yes, this algorithm could absolutely be used in such a setting, though note that we require that outcomes $Y$ are not causally influenced by the predictions $\hat{Y}$. This precludes the possibility that even observing the outcome is contingent on the human's decision, as is likely the case in e.g. college admissions. This is discussed in Section 6 and further in Appendix A.
> 2. With that caveat, we could certainly test whether human forecasts are independent of sensitive attributes (e.g., race) after conditioning on some set of `allowable' features. This would be very similar to the definition of conditional statistical parity given in "Algorithmic decision making and the cost of fairness'' (Corbett-Davies et. al, 2017). Alternatively, one could test whether the predictions $\hat{Y}$ and sensitive attributes are independent conditional on the outcome $Y$, as in "Equality of Opportunity in Supervised Learning'' (Hardt et. al, 2016).
> 3. Depending on the quantity of interest -- for example, whether forecasts are *unrelated* to race or whether they are systematically *lower* for a particular race -- one might consider modifying our test to be a two-tailed rather than single-tailed hypothesis test.
>
> **In response to**
> > "In Section 2, first paragraph: why can you assume X is independent of U, without a loss of generality?"
>
> This concern is addressed in our response to all reviewers (see above).
>
> **In response to**
> > "The results and experiments of this paper would likely not extend to a high-dimensional setting."
>
> This concern is addressed in our response to all reviewers (see above).
>
> **In response to**
> > "I understand it’s not an easy task, but this paper would be much stronger if there were additional experiments where this technique was employed to understand the interplay between human and algorithmic decision making."
>
> This concern is addressed in our response to all reviewers (see above).

---

> > ### Comment · Reviewer_7Ctt · 2023-08-16
> >
> > Thank you for your helpful responses! I have increased my review rating to a 7 due to the clarifications and the new results you posted above.

---

### Official Review · Reviewer_7cTX · 2023-07-04

**Soundness:** 3 good
**Presentation:** 3 good
**Contribution:** 3 good
**Rating:** 7
**Confidence:** 3

**Summary:**

This paper proposes a method for determining whether a human expert is usefully using outside information that a model does not incorporate in order to make decisions. The goal is to test whether complementarity, humans working with models, is possible for a given task. The paper sets forth an algorithm, ExpertTest, provides some theoretical guarantees, and uses emergency room admissions as a case study for the technique. In the case study, the method indicates that doctors are making use of external information that is not captured in a commonly used risk score.


**Strengths:**

The main strength of this paper is the novel problem that it seeks to solve. Understanding whether and how humans can add their expertise on top of automated decision systems is an important goal, and it is often understudied. It is a creative approach to a crucial problem.


**Weaknesses:**

The primary weakness of the paper, in my opinion, is that the problem setup focuses on variables, or features, rather than also considering the functional form for the prediction itself. Fundamentally, if we are comparing the performance of something like the Glasgow-Blandford score (GBS) to humans, how do we know that the human is not using the same exact input features $X$ as the GBS, but just has a better way of mapping those $X$ to the prediction we are interested in? Why does $U$ have to exist at all for the human to be lending their “expertise” to the problem? That is, let’s say the GBS output is $\tilde{Y} = \tilde{f}(X)$, and the human output is $\hat{Y} = \hat{f}(X)$. If $\hat{f}$ is closer to the actual generating function in the ground truth than $\tilde{f}$, then the human could be “adding expertise” without actually using additional information as defined in this paper. Perhaps the human’s training would help them map these variables better than the GBS algorithm does. If I am missing something here, please correct me, but it’s very unclear to me why the method proposed leads to the conclusion that the expert is using additional features to make a decision.


**Questions:**

1) Early in the paper, the authors equate $X$ with features, such as input features for a model. However, in the case study, the authors define $X$ as a GBS score, which is the output of a risk model. I think the notation needs to more carefully distinguish between what is an input to a model and what is an output of a model, and this may be part of the confusion that I express above in the weaknesses section. Could you clarify this point?

2) In the case study, $\hat{Y}$ is defined as whether the patient is admitted, and the ground truth $Y$ is defined as whether they suffered one of three adverse outcomes. Are these variables correlated? If a patient is admitted, are they less likely to suffer an adverse outcome? It seemed like this might actually be a setup where the “prediction” and “ground truth” are not independent of each other.

3) The third adverse outcome mentioned is initial discharge and readmission within 30 days. Does this mean that anyone who suffered that outcome had a $\hat{Y}$ of 0? I am just wondering if “initial discharge” means that the ER physician made the choice to not admit them.

4) How sensitive is ExpertTest to the choice of loss function?


**Limitations:**

I think the paper should more carefully address the assumptions in the initial problem setup, specifically what I lay out in the weaknesses section. What model of human expertise is being considered when the assumption is that any additional information is encapsulated in the $U$ variables?

---

> ### Author Rebuttal · Authors · 2023-08-08
>
> Thank you for your feedback! Our responses are provided below.
>
> **In response to**
> > "The primary weakness of the paper…is that the problem setup focuses on variables, or features, rather than also considering the functional form for the prediction itself."
>
> **We reproduced our original case study using the clinical features which make up the Glasgow-Blatchford Score. The results are attached with our response and discussed below.** We will include these results alongside our old ones in the final draft of Section 5. We respectfully disagree that the case study in our original draft is invalid however, and first provide a clarification:
>
> 1. Recall that we seek to test whether the predictions $\hat{Y}$ incorporate information other than the observable features $X$ to forecast an outcome $Y$.
> 2. In Section 5, we instantiate this framework where $X$ is *just* the Glasgow-Blatchford score (GBS). That the GBS is itself a summary of other clinical markers **is merely provided as context. Our goal is to test whether physicians incorporate information other than the GBS itself**.
> 3. You are absolutely correct that physicians may be relying on the *same* clinical markers but perhaps in a different way (i.e. by implicitly employing a different functional form). This is not a contradiction however -- the interpretation of ExpertTest simply depends on the set of features we choose to condition on.
> 4. We agree however that it is helpful to see the results both conditional on the GBS itself and conditional on its constituent parts. Indeed, these results strengthen our work significantly.
>
> **Description of ‘input’ features**
>
> We briefly contextualize the attached figures here, focusing on technical validity due to space constraints. Our final draft will include additional context regarding the clinical relevance of these features. The Glasgow-Blatchford Score is composed of the following nine features: blood urea nitrogen (BUN), hemoglobin (HGB), systolic blood pressure (SBP), pulse, cardiac failure, hepatic disease, melena, sycope and biological sex. The first four are continuous and the latter five are binary. The GBS is calculated by first converting the continuous features to ordinal values (BUN and HGB to 6 point scales, SBP to a 3 point scale and pulse to a binary value) and then summing the values of the first 8 features. Biological sex is used to inform the conversion of HGB to an ordinal value. We refer to this mix of binary and ordinal values as the *discretized* feature space. It is worth emphasizing that this score -- including the conventions used to convert the features to discrete space -- is part of the standard of care for patients with acute gastrointestinal bleeding, and has been extensively validated empirically (see Section 5 and Appendix E for additional details).
>
> **Summary of new results**
> We attach the results of running ExpertTest in both the original and discretized feature space. In both cases, we further normalize each feature to lie in $[0, 1]$ (to ensure that no feature is given outsize weight in calculating pairwise similarity). We find strong evidence that physicians incorporate information other than the 9 features described above when making hospitalization decisions ($\tau \approx 0$). In the original feature space, which includes continuous features, we naturally fail to find identical pairs of patients. Nonetheless, the pairs we find are very close under the $\ell_2$ norm, suggesting that ExpertTest remains approximately valid. In the discretized feature space we succeed in finding up to $L=1000$ pairs of identical patients, meaning that ExpertTest recovers (effectively) exact type I error control.
>
> **In response to**
> > In the case study, $\hat{Y}$ is defined as whether the patient is admitted, and the ground truth $Y$ is defined as whether they suffered one of three adverse outcomes…It seemed like this might actually be a setup where the “prediction” and “ground truth” are not independent of each other."
>
> 1. $Y$ and $\hat{Y}$ will typically be correlated (as they are here, and in any setting where the human makes reasonable forecasts). The condition we require (see Section 6 and Appendix A) is that $\hat{Y}$ does not *causally* influence $Y$.
> 2. We make this assumption here because (1) the ER physician making hospitalization decisions and GI specialist making treatment decisions are different physicians working in a large hospital system -- indeed, there are many cases where the GI specialist immediately discharges a patient admitted by the ER physician -- and (2) we assume we can still observe $Y=1$ outcomes for discharged patients ($\hat{Y} = 0$) who are readmitted or die within 30 days.
> 3. This latter assumption is necessarily imperfect, but is consistent with standard 'adverse outcome' definitions in the literature and Center for Medicare Services (CMS) guidelines. Relevant citations and further discussion of this outcome definition are included in Section 5 and Appendix E.
>
> **In response to**
> > "How sensitive is ExpertTest to the choice of loss function?"
>
> In the special case of binary outcomes and predictions (as in our case study), ExpertTest will provide identical results for any loss function which is strictly increasing in the number of prediction mistakes. This will be true of nearly any natural loss function. The results of ExpertTest also do not depend on the relative cost of false negatives and false positives, which may be arbitrarily different (e.g., false negatives in a clinical setting may be far more costly than false positives; we need not specify exactly *how much* more costly as our results are insensitive to this choice). **We discuss this phenomenon further in Appendix E.** In more general settings we also find that ExpertTest is also not very sensitive to the choice of loss function, and will include additional experiments to demonstrate this in our final submission (unfortunately we cannot do so here due to space constraints).

---

> > ### Comment · Reviewer_7cTX · 2023-08-16
> >
> > Thank you to the authors for their detailed response and additional experiments. My concerns about the features have been addressed and I have a better understanding of the method now. Given this response and the other comments, I am raising my score from 3 to 7.

---

### Official Review · Reviewer_DMEm · 2023-07-06

**Soundness:** 3 good
**Presentation:** 3 good
**Contribution:** 3 good
**Rating:** 7
**Confidence:** 4

**Summary:**

This paper proposes a hypothesis-testing approach to identify whether a set of predictions made by a human expert uses additional information that is conditionally independent from the input covariates. The paper provides theoretical guarantees for the test in a general setting and asymptotically. The proposed test is then applied to real-world data with emergency room physicians.

**Strengths:**

- The paper itself is written in a way that is easy to follow along, building the motivation and proposed work in a step-by-step manner.
- The paper proposes a test that is a novel application of conditional independence work in the statistics literature for a novel use case.
- The paper evaluates the theoretical test on a real-world use case.


**Weaknesses:**

- Can the test help improve decision outcomes? Typically, the primary goal in human-AI decision-making is to achieve complementarity (e.g., as discussed in [1]), particularly by leveraging the complementary skills of human and AI. Because the test does not account for human and AI prediction accuracies, it is difficult to say whether performing such a test has any implications on complementarity. Another relevant cite is [2].

[1] Does the Whole Exceed its Parts? The Effect of AI Explanations on Complementary Team Performance. Bansal et al. CHI 2021.

[2] A Unifying Framework for Combining Complementary Strengths of Humans and ML toward Better Predictive Decision-Making. Rastogi et al. EAAMO 2022.

- Experimental validation: While it is great that the authors perform experiments on real-world data, it would be ideal to also verify the behavior of the test using synthetic where the differences between human and AI can be more carefully controlled to establish how sensitive the test is to these similarities and potentially the effect on the choice of L.


**Questions:**

Major:
- Could you address the questions raised in the weaknesses mentioned above?

Minor:
- Notation confusion: U is used to both denote private information in Section 2 and random variables and observations in Section 3
- What would the test return if the human is using information U that is *correlated* with X? Here, this may be a case where U is easier for a human to notice compared to X.
- The table captions are not descriptive, which hinders readability.
- Tone of contribution: the authors state that this test is a “necessary condition for achieving human-AI complementarity” without demonstrating results in terms of improving decision outcomes.


**Limitations:**

Yes

---

> ### Author Rebuttal · Authors · 2023-08-08
>
> Thank you for your feedback! Our responses are provided below.
>
> **In response to**
> > "Can the test help improve decision outcomes? Typically, the primary goal in human-AI decision-making is to achieve complementarity (e.g., as discussed in [1]), particularly by leveraging the complementary skills of human and AI. Because the test does not account for human and AI prediction accuracies, it is difficult to say whether performing such a test has any implications on complementarity. Another relevant cite is [2].".
>
> 1. This is an excellent question, and is addressed in Section 6 -- while the algorithm we give here is focused on the auditing problem, we hope that our framework yields insight into the structure of algorithms for Human/AI complementarity.
> 2. While providing such algorithms is beyond the scope of our work, we respectfully disagree that "it is difficult to say whether performing such a test has any implications on complementarity". A rejection of our test indicates that the human forecaster *is* usefully incorporating information which is unavailable to any predictive algorithm, even if such information is incorporated in a suboptimal way.
> 3. This suggests that, for example, *some* meta-algorithm which incorporates both human and baseline algorithmic forecasts should be able to achieve complementarity (given sufficient data, expressive power and other considerations which are standard in supervised learning tasks). On the other hand, a failure to reject our test suggests that *no* algorithm could hope to achieve complementarity, as the human is not incorporating useful information beyond that which is contained in the available features.
> 4. Thus, we view our test as a precursor to assessing whether complementarity is likely to be achievable in a given forecasting task.
>
> **In response to**
> > "Experimental validation: While it is great that the authors perform experiments on real-world data, it would be ideal to also verify the behavior of the test using synthetic where the differences between human and AI can be more carefully controlled to establish how sensitive the test is to these similarities and potentially the effect on the choice of L."
>
> This concern is addressed in our response to all reviewers (see above).
>
> **In response to**
> > "What would the test return if the human is using information U that is correlated with X? Here, this may be a case where U is easier for a human to notice compared to X."
>
> This concern is addressed in our response to all reviewers (see above).
>
> **In response to other feedback**
>
> We agree with the remaining suggestions, and thank you for your feedback. We will incorporate the two suggested references and change the notation in Section 3 to avoid clashing with Section 2.

---

> > ### Comment · Reviewer_DMEm · 2023-08-16
> >
> > Thank you to the authors for clarifying my concern about how this work should be situated in the broader space of human-AI decision-making. As the authors suggested in their rebuttal, I agree that a few minor proposed changes (highlighting the synthetic experiments and cleaning up notation / presentation) would certainly improve the work. As such, I will modify my score from 6 to 7.

---

### Author Rebuttal · Authors · 2023-08-08

We thank all four of the reviewers for their thoughtful and constructive feedback. We address feedback provided by more than one reviewer in this comment, with additional responses to individual reviewer concerns provided in-line below each review.

# High dimensional feature spaces
> **7Ctt:** "The results and experiments of this paper would likely not extend to a high-dimensional setting. Their current experiment uses discrete, scalar features. The authors discuss this in their limitations. A concrete example: how would a test like this work for, say, radiology images, where the human predictive distribution is likely not smooth w.r.t the $\ell_2$ metric."

> **NoBC:** "Method relies on dataset containing pairs that are close in input space, yet distinct in feature space; such a situation might not be generalizable"


1. Per the proof of Theorem 2 (Appendix C), the excess type I error scales like $O(n^{-\frac{1}{d}})$ for a $d$ dimensional feature space. This behavior is typical in a generic, non-parametric setting (see e.g. the model-powered test of conditional independence of Sen et. al, 2017, which recovers the same bound).
2. If there were a *latent* lower dimensional representation, as is often the case with e.g., image data, our results could potentially be adapted to be with respect to the latent lower dimension. Indeed, as **7Ctt** points out, the smoothness condition required for Theorem 2 may not hold in a high dimensional setting, but may hold with respect to the latent representation.
3. It is also worth noting that these are *worst case* results over the set of possible distributions on $X, Y, \hat{Y}$. We find in both our real-world case study (Section 5 and figures attached to this rebuttal) and synthetic experiments (Appendix F) that typical distributions are better behaved; in particular, they exhibit more 'clustering' of similar $(x_i, x_j)$ pairs and thus require smaller datasets to achieve acceptable type I error.
4. We discuss alternative tests (e.g., kernel based tests of conditional independence) which may be better suited for high-dimensional feature spaces in Section 6.


# Additional experiments, synthetic data
> **DMEm**: "Experimental validation: While it is great that the authors perform experiments on real-world data, it would be ideal to also verify the behavior of the test using synthetic where the differences between human and AI can be more carefully controlled to establish how sensitive the test is to these similarities and potentially the effect on the choice of L."

> **NoBC**: "Evaluation is only done on one real-world dataset; a controlled evaluation of the test would give better insights into how the test performs and the impact of different parameters"

1. Our manuscript includes extensive experiments on synthetic data in Appendix F, which is included with our original set of supplementary material. It is likely that the reviewers missed this section.
2. We reference these experiments in Section 1, and will feature them more prominently in our edited draft.


# On the potential correlation of $X$ and $U$
> **DMEm:** "What would the test return if the human is using information U that is correlated with X? Here, this may be a case where $U$ is easier for a human to notice compared to $X$."

> **7Ctt:** "In Section 2, first paragraph: why can you assume X is independent of U, without a loss of generality?"

1. First, **there is indeed a typo in our work**; we intended to state that $X$ and $U$ can be assumed to be *uncorrelated* without loss of generality, not independent.
2. Neither assumption (independence or zero correlation) is used anywhere in our analysis however; this sentence was merely intended to clarify our framework.
3. Zero correlation can be assumed without loss of generality because we can take $U' = (U - E[U \mid X])$ and replace $U$ by $U'$. Thus, even if the human forecaster incorporates unobserved information which is correlated with $X$, we can define $U$ to be the *additional* information which is not correlated with the observed features.
4. **Given the confusion around this point, our proposal is to simply delete this sentence from the final draft.** If the reviewers prefer, we are also happy to change 'independent' to 'uncorrelated' and clarify this point in the final draft.


# Formatting of tables
> **DMEm:** "The table captions are not descriptive, which hinders readability."

> **NoBC:** "In table 2, what does mismatched pairs mean?"

This is excellent feedback, and we will include more descriptive captions in the final draft. We answer NoBC's specific question in our individual response below.

---

### Decision · Program_Chairs · 2023-09-21

**Decision:**

Accept (spotlight)

**Comment:**

The reviewers all agree on the contribution and novelty. Thus I recommend acceptance.